# Identification of Differential Compositions of Aqueous Extracts of Cinnamomi Ramulus and Cinnamomi Cortex

**DOI:** 10.3390/molecules28052015

**Published:** 2023-02-21

**Authors:** Pei Wang, Jun Chi, Hui Guo, Shun-Xiang Wang, Jing Wang, Er-Ping Xu, Li-Ping Dai, Zhi-Min Wang

**Affiliations:** 1Henan University of Chinese Medicine, Zhengzhou 450046, China; 2Engineering Technology Research Center for Comprehensive Development and Utilization of Authentic Medicinal Materials in Henan Province, Henan University of Chinese Medicine, Zhengzhou 450046, China; 3National Engineering Laboratory for Quality Control Technology of Chinese Herbal Medicines, Institute of Chinese Materia Medica, China Academy of Chinese Medical Sciences, Beijing 100700, China

**Keywords:** Cinnamomi ramulus, Cinnamomi cortex, LC-MS, multivariate statistical analysis, quantitative analysis, molecular docking

## Abstract

Cinnamomi ramulus (CR) and Cinnamomi cortex (CC), both sourced from *Cinnamomum cassia* Presl, are commonly used Chinese medicines in the Chinese Pharmacopeia. However, while CR functions to dissipate cold and to resolve external problems of the body, CC functions to warm the internal organs. To clarify the material basis of these different functions and clinical effects, a simple and reliable UPLC-Orbitrap-Exploris-120-MS/MS method combined with multivariate statistical analyses was established in this study with the aim of exploring the difference in chemical compositions of aqueous extracts of CR and CC. As the results indicated, a total of 58 compounds was identified, including nine flavonoids, 23 phenylpropanoids and phenolic acids, two coumarins, four lignans, four terpenoids, 11 organic acids and five other components. Of these compounds, 26 significant differential compounds were identified statistically including six unique components in CR and four unique components in CC. Additionally, a robust HPLC method combined with hierarchical clustering analysis (HCA) was developed to simultaneously determine the concentrations and differentiating capacities of five major active ingredients in CR and CC: coumarin, cinnamyl alcohol, cinnamic acid, 2-methoxycinnamic acid and cinnamaldehyde. The HCA results showed that these five components could be used as markers for successfully distinguishing CR and CC. Finally, molecular docking analyses were conducted to obtain the affinities between each of the abovementioned 26 differential components, focusing on targets involved in diabetes peripheral neuropathy (DPN). The results indicated that the special and high-concentration components in CR showed high docking scores of affinities with targets such as HbA1c and proteins in the AMPK–PGC1–SIRT3 signaling pathway, suggesting that CR has greater potential than CC for treating DPN.

## 1. Introduction

Cinnamomi ramulus (CR, *Guizhi* in Chinese) and Cinnamomi cortex (CC, *Rougui* in Chinese), are, respectively, the dried twigs and bark of *Cinnamomum cassia* Presl. They are documented in the Chinese Pharmacopoeia, which has a long history and comprises all well-known Chinese medicines with a high clinical value [1]. CR functions to dissipate wind-cold and resolve external problems of the body; it traverses the arms and warms the meridians. It is known as the first medicine of classic prescription and is widely used in the Treatise on Cold Pathogenic and Miscellaneous Diseases [2]. In contrast, CC functions to warm the internal organs, nourish and warm kidney-yang, dispel cold, and relieve pain. Different pharmacological effects result from variant chemical ingredients. It can be assumed that several significant components differ in CR and CC, leading to differences in their respective efficacies. Previous chemical investigations demonstrated that CR and CC contained various compositions, including phenylpropanoids, phenolic acids, flavonoids and terpenoids and so on [3,4,5]. A previous study showed that the contents of volatile oils in CR and CC, were 0.15% and 0.31%, respectively, with the main component being cinnamaldehyde [6]. The traditional usage of decocting with water requires that it is necessary to continue to analyze the different components of the aqueous extracts of CR and CC.

In this study, a simple, rapid, valid, and reliable UPLC-Orbitrap-Exploris-120- MS/MS [7,8,9,10] method was developed to assess the difference in the chemical compositions of CR and CC. By comparing the MS/MS information of the detected compounds with the mass spectrometry database, records in the literature, and standard references, 58 compounds were preliminary identified. Eight batches of CR and CC samples were comparatively analyzed using this method coupled with multivariate statistical analysis [11,12,13]. As a result, 26 statistically significant (*p* < 0.05) discrepant compounds were characterized. Additionally, a robust HPLC method combined with hierarchical clustering analysis (HCA) was developed to simultaneously determine the concentrations and differentiating capacities of five major active ingredients in CR and CC: coumarin, cinnamyl alcohol, cinnamic acid, 2-methoxycinnamic acid and cinnamaldehyde. The HCA results showed that *trans*-cinnamaldehyde and cinnamyl alcohol could be used as markers to successfully distinguish CR and CC.

Diabetes peripheral neuropathy (DPN), one of the most common complications of diabetes mellitus, is commonly categorized as Xiaoke (diabetes) complicated by arthralgia syndrome in TCM [14]. As a diaphoretic, CR functions to dissipate wind-cold and is usually used for treating arthralgia syndrome caused by rheumatoid arthritis and diabetes [15,16]. CR is always included in TCM formulars for treating DPN, e.g., *Huangqi Guizhi Wuwu* Decoction [17,18]. In clinics, HbA1c is used as an essential indicator to evaluate the severity of DPN [19,20]. As a vital signaling pathway, AMPK–PGC1–SIRT3 was widely reported and involved in the occurrence and development of DPN [21]. Using virtual molecular docking technology, we, therefore, explored whether any ingredients occurring in CR are different from those in CC and, if so, whether they contribute to anti-DNP activities. The findings of this study will provide comparative information on the chemical profiles of CR and CC and lay the groundwork for exploring effective CR substances as anti-DNP agents.

## 2. Results and Discussion

### 2.1. Optimization of the LC-MS Conditions

In order to improve the sensitivity and resolution of the analysis but reduce analytical time, the LC-MS conditions for aqueous extract of CR, which included mobile phase, flow rate, column temperature, ion mode and ion source parameters, were optimized. First, methanol/0.1% formic acid water and acetonitrile/0.1% formic acid water were investigated to achieve better separation. It was found that acetonitrile/0.1% formic acid water was the most suitable to obtain more peak and better peak shape. Second, we compared two types of columns (Waters Acquity UPLC BEH C18 and HYPERSILGOLD Vanquish C18) and found that the latter column had better resolution and more peaks. Different column temperatures (25 and 35 °C) were tested and a good separation effect and higher peak intensity was obtained for the constituents at 35 °C. In addition, we also compared the vaporizer temperature (320 and 350 °C) and the positive ion spray voltage (3.8 and 3.5 kV), and finally we found that the TIC had high sensitivity and good resolution when the vaporizer temperature was 350 °C and the positive ion spray voltage was 3.5 kV. The above data are shown in Appendix A.

### 2.2. Identification of the Constituents of CR and CC

Qualitative analysis of the chemical constituents of the aqueous extracts of CR and CC was performed using UPLC-Oribtrap-Exploris-120-MS/MS, in both in the positive and negative modes. In the qualitative analysis, in order to evaluate the stability of the instrument, mixed standards (QC) were repeated for 8 times. The RSD_S_ of the retention times and intensities in QC samples were all less than 5%. Typical total ion chromatograms for CR and CC, both in negative ion mode are shown in Appendix A. A comparison of retention times, accurate mass, fragmentation patterns, and a comparison with an online database, as well as referencing to the related literature to validate the data, preliminarily identified 58 compounds (Table 1, Appendix A); these included nine flavonoids, 23 phenylpropanoids and phenolic acids, two coumarins, four lignans, four terpenoids, 11 organic acids and five other components. Among these compounds, eight compounds were identified by comparing their retention times, characteristic molecular ions, and fragment ions to those of the standards (Appendix A). Significantly, among these compounds, 26 significant differential compounds were identified statistically: the results showed that the peak areas of flavonoids and flavonoid glycosides, and phenylpropanoids except cinnamaldehyde and cinnamyl alcohol in CR were higher than in CC, and the peak areas of terpenoids and organic acids in CC were higher than in CR.

#### 2.2.1. Identification of Phenylpropanoids and Phenolic Acids

Phenylpropanoids and phenolic acids are major bioactive constituents in CR and CC [22]. *Trans*-cinnamic acid, *trans*-cinnamaldehyde, cinnamyl alcohol and 2-methoxycinnamic acid were shown to be the main phenylpropanoids. In the literature, the diagnostic ions at *m*/*z* 163/165, 151/153 and 135/137 105/107 were indicators of phenylpropanoids and phenolic acids [23]. However, in addition to the above diagnostic ions reported in the literature and standards, our statistical analysis found that fragment ions at *m*/*z* 105/107 and 123/125 can also be diagnostic ions [23].

With the transfer of electrons, the carbonyl group was also prone to cleavage and the loss of CO and CH_2_ to form fragment ion peaks [24,25]. As a result, when compared to the diagnostic ions described above, the mass spectra of the standards, fragmentation information in the database and previously reported fragmentation spectra, a total of 23 phenylpropanoids and phenolic acids were identified in CR and CC. Compounds **45** and **48** were chosen as typical examples to illustrate the fragmentation pathways.

The molecular ion peak *m*/*z* 149.0231 [M + H]^+^ was obtained in the ESI (+) mode for compound **45**, which had the molecular formula C_9_H_8_O_2_ and a relative molecular mass of 148.0525. The matching fragments were *m*/*z* 131.0493 [M − H_2_O + H]^+^ and *m*/*z* 105.0539 [M − CO + H]^+^ (Table 1), which were in general agreement with the literature and data available for cinnamic acid; accordingly, the compound was presumed to be *trans*-cinnamic acid [26]. The MS^2^ spectrum of compound **45** is shown in Appendix A and the possible cleavage process of the positive ions is shown in Figure 1.

**Table 1 molecules-28-02015-t001:** Identification of chemical constituents in CR and CC.

NO.	Name	RT	Formula	Calc. MW	Error (ppm)	Theoretical Mass (*m*/*z*)	Experimental Mass (*m*/*z*)	MS^2^ (*m*/*z*)	Total Score (%) ^a^	Ref.	Source ^b^
**1**	Gentisic acid-5-O-glucoside	3.15	C_13_H_16_O_9_	316.0781	−0.50	315.0711 [M − H]^−^	315.0709 [M − H]^−^	270.8695 [M − COOH + H]^+^ 165.0183, ***153.0185*** [M − C_6_H_10_O_5_ + H]^+^, 152.0108, 113.0240, 109.0293 [M − C_6_H_10_O_5_ − COOH + H]^+^, 108.0211	91.2	[27]	
**2**	Isovanillic acid	3.22	C_8_H_8_O_4_	168.0415	1.88	167.0339 [M − H]^−^	167.0342 [M − H]^−^	152.0116 [M − CH_3_ − H]^−^, ***151.0226*** [M − OH]^−^, 123.0449 [M − COOH − H]^−^, 108.0217 [M − COOH − CH − H]^−^		[28,29,30]	CR, *C. cassia* leaves
**3**	Gentisic acid	3.30	C_7_H_6_O_4_	154.0257	3.04	153.0182 [M − H]^−^	153.0187 [M − H]^−^	***153.0187***, 135.0182 [M − H_2_O + H]^+^, 109.0288 [M − COOH + H]^+^, 85.0289, 81.0343 [M − COOH − CO + H]^+^, 68.9978	96.6	[31,32]	*C. cassia*
**4**	Syringic acid	3.65	C_9_H_10_O_5_	198.0527	−4.53	199.0601 [M + H]^+^	199.0588 [M + H]^+^	181.0495 [M − OH + H]^+^, 163.1478 [M − 2H_2_O + H]^+^, 155.0166, ***153.0764*** [M − OH − OCH_3_ + H]^+^, 95.0492	75.3	[31]	CR, CC, *C. cassia* leaves
**5**	Catechol	3.80	C_6_H_6_O_2_	110.0363	4.05	109.0284 [M − H]^−^	109.0291 [M − H]^−^	108.0215 [M − 2H]^−^, 93.7792 [M − OH − H]^−^, 81.6772	97.4	[33]	CR
**6**	Neochlorogenic acid	4.40	C_16_H_18_O_9_	354.0936	−4.21	353.0892 [M − H]^−^	353.0886 [M − H]^−^	191.0550 [M − C_9_H_6_O_3_ − H]^−^, 179.0343 [M − C_7_H_10_O_5_ − H]^−^, 161.0237 [M − C_7_H_10_O_5_ − H_2_O − H]^−^, 135.0445 [M − C_7_H_10_O_5_ − CO_2_ − H]^−^, 111.0446		[34]	CR, CC
**7**	Salicylic acid	5.38	C_7_H_6_O_3_	138.0311	3.50	137.0233 [M − H]^−^	137.0238 [M − H]^−^	119.0132 [M − H_2_O − H]^−^, 108.8992 [M − CO − H]^−^, 93.0343 [M − COO − H]^−^	62.9	[31,35]	CR, CC
**8**	Citrinin	5.40	C_13_H_14_O_5_	250.0840	−0.46	251.0916 [M + H]^+^	251.0915 [M + H]^+^	233.0804 [M − H_2_O + H]^+^, 221.0807 [M − 2CH_3_ + H]^+^, 205.0857 [M − H_2_O − CO + H]^+^, 204.0785, 191.0701	75.5		
**9**	4-Methoxy benzaldehyde	6.30	C_8_H_8_O_2_	136.0523	−0.77	137.0597 [M + H]^+^	137.0596 [M + H]^+^	122.0362 [M − CH_3_ + H]^+^, ***107.0490*** [M − OCH_2_ + H]^+^, 93.0590 [M − CO *−* CH_3_ + H]^+^, 91.0543, 81.0698, 79.0543 [M − CO − OCH_3_ + H]^+^	81.9	[28,36]	CR, CC, *C. cassia* leaves
**10**	Darendoside A	6.32	C_19_H_28_O_11_	432.1613	−2.29	431.1547 [M − H]^−^	431.1538 [M − H]^−^	191.0547, 161.0446, 149.0447,113.0245, 99.0081, 89.0244	67.5		
**11**	Epicatechin	8.41	C_15_H_14_O_6_	290.0788	−1.08	291.0863 [M + H]^+^	291.0860 [M + H]^+^	**273.0768** [M − H_2_O + H]^+^, **249.0766**, **179.0340** [M−C_6_H_6_O_2_+H] ^+^, **165.0544** [M − C_6_H_6_O_2_ − CH_2_ + H]^+^, 151.0394, **139.0388** [M − C_9_H_11_O_2_ + H]^+^,125.0239 [M − C_9_H_8_O_3_ + H]^+^, **123.0439**, 119.0491, 109.0290 [M − C_9_H_10_O_4_ + H]^+^, **95.0490**	99.4	Standard, [37]	CR
**12**	2,4,6-Trihydroxy-2-(4-hydroxybenzyl)-1-benzofuran-3(2H)-one	8.68	C_15_H_12_O_6_	288.0621	−4.34	287.0521 [M − H]^−^	287.0511 [M − H]^−^	161.0233 [M − C_6_H_6_O_3_ − H]^−^, 131.2500, 125.0239 [M − C_8_H_2_O_4_ − H]^−^	88.7		
**13**	Dihydrophaseic acid	8.75	C_15_H_22_O_5_	282.1454	−0.53	281.1383 [M − H]^−^	281.1382 [M − H]^−^	237.1486 [M − CHO − H]^−^, 201.1273 [M − CHO − H_2_O − OH − H]^−^, 189.1278, 171.1171, 139.0758	76.8		
**14**	Catechin	8.83	C_15_H_14_O_6_	290.0777	−1.26	***289.0707*** [M − H]^−^	***289.0703*** [M − H]^−^	***289.0703*** [M − H_2_O − H]^−^, 271.0603, 245.08081, 203.07036, 151.0393, 125.0239, 109.0290		[35]	CC
**15**	Vanillin	9.28	C_8_H_8_O_3_	152.0472	−0.82	***153.0548*** [M + H]^+^	***153.0545*** [M + H]^+^	***153.0545***, 125.0595 [M − CO + H]^+^, 111.0441, 93.0333, 65.0387	90.7	[28,31,38]	*C. cassia* leaves
**16**	4-Acetyl-3-hydroxy-5-methylphenyl-*β*-D-glucopyranoside	10.08	C_15_H_20_O_8_	328.1144	−1.05	327.1074 [M − H]^−^	327.1071 [M − H]^−^	165.0547 [M − C_6_H_10_O_5_ − H]^−^, 147.0446 [M − C_6_H_10_O_5_ − H_2_O − H]^−^, 121.0653 [M − C_6_H_10_O_5_ − OH − CH_3_ − CH_2_ − OH]^−^, 106.0416	91.7	[27]	
**17**	Picconioside B	10.95	C_26_H_40_O_12_	544.2500	−3.66	543.2438 [M − H]^−^	543.2421 [M − H]^−^	525.2305 [M − H_2_O − H]^−^, 363.1800 [M − H_2_O − C_6_H_10_O_5_ − H]^−^, 381.1912, 167.1070, 165.0922, 101.0240, 89.0240, 59.0136	91.2		
**18**	2 Methoxybenzoic acid	12.68	C_8_H_8_O_3_	152.0472	−0.92	***153.0546*** [M + H]^+^	***153.0545*** [M + H]^+^	***153.0545***, 135.0439, 111.0440, 105.0441, 95.0491 [M − CO_2_ − CH_2_ + H]^+^, 93.0699, 79.0541	96.3	[39,40]	CR
**19**	Taxifolin	13.79	C_15_H_12_O_7_	304.0581	2.03	305.0656 [M + H]^+^	305.0662 [M + H]^+^	***287.0573*** [M − H_2_O + H]^+^, **259.0591** [M − CO − H_2_O + H]^+^, **231.0652**, **153.0188** [M − CO − C_7_H_8_O_2_ + H]^+^, **149.0230**	91.1	Standard, [12,41]	CR
**20**	Lyoniresinol-3a-O-*β*-D-glucopyranosid	13.89	C_28_H_38_O_13_	582.2289	−4.09	581.2230 [M − H]^−^	581.2209 [M − H]^−^	566.1975, 535.1785, 419.1691 [M − C_6_H_10_O_5_ − H]^−^, 404.1459, 373.1275 [M − C_6_H_10_O_5_ − 3CH_3_ − H]^−^, 359.1110 [M − C_6_H_10_O_5_ − 4CH_3_ − H]^−^, 233.0812, 202.0624, 153.0549, 138.0316 [M − C_6_H_10_O_5_ − OH − C_14_H_18_O_6_ − H]^−^, 101.0238	92.3	[33,42,43]	CC
**21**	4-Ethylphenol	14.12	C_8_H_10_O	122.0726	4.32	121.0649 [M − H]^−^	121.0654 [M − H]^−^	106.0419 [M − CH_3_ − H]^−^, 90.9232 [M − CH_3_ − O − H]^−^, 61.9880	87.9	[44,45]	*C. cassia*
**22**	(−)-Lyoniresinol	14.19	C_22_H_28_O_8_	420.1766	−1.53	419.1700 [M − H]^−^	419.1694 [M − H]^−^	373.1277 [M − 3CH_3_ − H]^−^,359.1119 [M −4 CH_3_ − H]^−^, 313.0712, 221.0801, 180.0404, 139.0396, 134.0383	96.0	[33]	
**23**	Lyoniside	14.33	C_27_H_36_O_12_	552.2186	−3.79	551.2123 [M − H]^−^	551.2105 [M − H]^−^	536.1875, 419.1650, 389.1591, 374.1359, 373.1275 [M − C_6_H_10_O_5_ − 3CH_3_ − H]^−^, 359.1105 [M−C_6_H_10_O_5_−4CH_3_−H]^−^, 341.1013, 325.1092, 233.0823, 119.0345, 113.0239	91.8	[33]	
**24**	3-Oxoindane-1-carboxylic acid	14.41	C_10_H_8_O_3_	176.0472	−0.69	177.0546 [M + H]^+^	177.0545 [M + H]^+^	153.9367, 149.0596 [M−CO+H] ^+^, 133.0646 [M−COO+H] ^+^, 131.0490, 121.1010, 107.0490, 105.0693 [M − CO − COO + H]^+^, 93.0098 [M − CO − COO − C + H]^+^, 81.0700 [M − CO − COO − 2C + H]^+^	72.2		
**25**	3-Methoxy phenylacetic acid	14.52	C_9_H_10_O_3_	166.0622	1.69	165.0546 [M − H]^−^	165.0549 [M − H]^−^	147.0443, 136.9315 [M − OCH_3_ − H]^−^, 121.0654, 106.0419, 96.9597 [M − C_2_H_3_O_2_ − H]^−^		[39]	
**26**	2-(4-Hydroxyphenyl)-7-((3,4,5-trihydroxy-6-(hydroxymethyl) tetrahydro-2H-pyran-2-yl) oxy) chroman-4-one	14.53	C_21_H_22_O_9_	418.1248	−0.74	417.1180 [M − H]^−^	417.1177 [M − H]^−^	301.0338 [M − C_6_H_10_O_5_ − H]^−^, 255.0651, 153.0187 [M − C_6_H_10_O_5_ − CO − C_7_H_6_O_2_ − H]^−^, 135.0082 [M − C_6_H_10_O_5_ − CO − C_7_H_6_O_2_ − H_2_O − H]^−^, 119.0497, 91.0184	67.5	[12]	
**27**	Quercetin-3*β*-D-glucoside	14.55	C_21_H_20_O_12_	464.0936	0.21	463.0871 [M − H]^−^	463.0872 [M − H]^−^	***301.0338*** [M − H − C_6_H_10_O_5_]^−^, 300.0270, 271.0247		[34]	
**28**	2-[1-(2H-1,3-Benzodiox ol-5-yl) propan-2-yl]-6-metho xy-4-(prop-2-en-1-yl) phenol	14.74	C_20_H_22_O_4_	326.1516	0.63	327.1590 [M + H]^+^	327.1593 [M + H]^+^	312.1348, 295.1328 [M − OCH_2_ + H]^+^, 280.1095, 263.1071, 251.0001, 235.1122, 175.0758, 163.0753 [M − C_10_H_12_O_2_ + H]^+^, 151.075, 137.0596, 133.0647, 103.0540, 98.9841	71.6		
**29**	Cinnamylalcohol-6′-O-*α*-furanara- binose-O-*β*-glucopyranoside	14.81	C_20_H_28_O_10_	428.1664	−4.38	427.1598 [M − H]^−^	427.1582 [M − H]^−^	293.0861, 233.0650, 191.0549, 161.0451, 149.0448, 125.0239, 89.0240, 81.0344, 59.0136	85.9	[46]	CR
**30**	6-Methoxymellein	14.99	C_11_H_12_O_4_	208.0734	−0.55	209.0804 [M + H]^+^	209.0802 [M + H]^+^	191.0701, 181.0847, 177.0544, 163.0765, 149.0596 [M − COOCH + H]^+^, 131.0486, 121.0647 [M − 2OH − OCH_3_ − CO + H]^+^, 109.0647, 103.0540, 93.0698, 91.0540, 55.0177	85.8		.
**31**	Coumarin	15.18	C_9_H_6_O_2_	146.0366	−1.33	147.0441 [M + H]^+^	147.0439 [M + H]^+^	**127.0543**, **103.0541** [M − CO_2_ + H]^+^, **91.0540** [M − 2CO + H]^+^, **43.0242**	96.8	Standard, [46]	CR, CC, *C. cassia* leaves
**32**	Quercetin	15.36	C_15_H_10_O_7_	302.0424	−0.98	***303.0499*** [M + H]^+^	***303.0496*** [M + H]^+^	***303.0496***, 275.0399 [M − CO + H]^+^, **257.0446** [M − CO − H_2_O + H]^+^, 247.0590, **229.0491** [M − 2CO − H_2_O + H]^+^, 199.0434 [M − 3CO − H_2_O + H]^+^, **165.0178**, 163.0389, **153.0183** [M − CO − C_7_H_6_O_2_ + H]^+^, 133.0231 [M − CO − C_7_H_4_O_2_ − H_2_O + H]^+^, **121.0297**, **111.0075**	99.8	[29,35]	CR
**33**	Quercitrin	15.36	C_21_H_20_O_11_	448.1003	−0.58	449.1089 [M + H]^+^	449.1087 [M + H]^+^	431.0983, 369.0594, 345.0606, 315.0494, 303.0497 [M + H − C_6_H_9_O_4_]^+^, 257.0439 [M − C_6_H_9_O_4_ − CO − H_2_O + H] ^+^, 229.0492 [M − C_6_H_9_O_4_ − 2CO − H_2_O + H]^+^, 129.0548, 85.0283, 71.0490	82.8	Standard	
**34**	Graveobioside A	15.36	C_26_H_28_O_15_	580.1402	−4.46	579.1344 [M − H]^−^	579.1325 [M − H]^−^	476.1080, 417.1531 [M − C_6_H_10_O_5_ − H]^−^, 300.0259 [M − C_6_H_8_O_4_ − C_5_H_10_O_4_ − H]^−^, 271.0235 [M − C_11_H_16_O_10_ − H]^−^, 178.9979 [M − C_17_H_20_O_11_ − H]^−^	85.3		
**35**	Libertellenone B	15.74	C_20_H_26_O_4_	330.1830	−0.32	331.1903 [M + H]^+^	331.1902 [M + H]^+^	313.1796 [M − H_2_O + H]^+^, 295.1676, 271.1686, 243.1763, 165.0911, 125.0565 [M − C_12_H_14_O_3_ + H]^+^	75.2		
**36**	Yucalexin P-17	16.06	C_17_H_20_O_3_	272.1412	0.27	273.1485 [M + H]^+^	273.1488 [M + H]^+^	255.1391 [M − H_2_O + H]^+^, 245.1534, 227.1441 [M − H_2_O − CO + H]^+^, 203.1070, 149.0964 [M − CH_3_ − C_6_H_6_O_2_ + H] ^+^, 82.8045 [M − C_11_H_10_O_3_ + H]^+^	86.6		
**37**	Azelaic acid	16.24	C_9_H_16_O_4_	188.1040	0.12	187.0965 [M − H]^−^	187.0966 [M − H]^−^	169.0861, 143.1072, 125.0966 [M − COOH − OH − H]^−^, 123.0811, 97.0654 [M − 2COOH − H]^−^, 57.0343	95.4	[15,47]	CR
**38**	Kaempferol	16.27	C_15_H_10_O_6_	286.0475	−0.89	287.0550 [M + H]^+^	287.0548 [M + H]^+^	***287.0548***, 258.0511 [M − CO + H]^+^, 183.0288, 165.0183, 153.0189 [M − CO − C_7_H_6_O + H]^+^, 133.0292,121.0281	99.2	[35]	CR, CC, *C. cassia* leaves
**39**	1-(Carboxymethyl) cyclohexane carboxylic acid	16.36	C_9_H_14_O_4_	186.0884	1.06	185.0808 [M − H]^−^	185.0819 [M − H]^−^	141.0916 [M − COO − H]^−^, 104.0775 [M − C_6_H_9_ − H]^−^	87.2		
**40**	Kaempferol-3-O-*α*-L-arabinopyranosyl-7-O-*α*-L-rhamnopyranoside	16.64	C_26_H_28_O_14_	564.1456	−4.12	563.1393 [M − H]^−^	563.1384 [M − H]^−^	435.2045, ***285.0416*** [M − C_6_H_8_O_4_ − C_5_H_10_O_4_ − H]^−^, 284.0316, 255.0286, 147.5166, 70.7867		[48]	
**41**	2-Methoxy benzaldehyde	16.67	C_8_H_8_O_2_	136.0522	−0.16	137.0597 [M + H]^+^	137.0596 [M + H]^+^	109.0647, ***107.0490*** [M − OCH_2_ + H]^+^, 93.0698 [M − CO − CH_3_ + H]^+^, 81.0697, 79.0512 [M − CO − OCH_3_ + H]^+^	90.9	[28]	CR, *C. cassia* leaves
**42**	Cinnamyl alcohol	17.22	C_9_H_10_O	134.0726	−0.89	**117.0698** [M + H − H_2_O]^+^	**117.0696** [M + H − H_2_O]^+^	**117.0696** [M − H_2_O + H]^+^, ***91.0540*** [M − C_2_H_4_O + H] 78.2648 [M − C_3_H_5_O + H]^+^, 63.4672, 49.4958		standard	CR, CC, *C. cassia* leaves
**43**	4-Methylumbelliferyl-*α*-D-glucopyranoside	17.37	C_16_H_18_O_8_	338.1002	0.21	339.1074 [M + H]^+^	339.1075 [M + H]^+^	321.0970 [M − OH + H]^+^, 177.0546 [M − C_6_H_10_O_5_ + H]^+^, 145.0284 [M − C_6_H_10_O_5_ − CH_3_ − OH + H]^+^, 127.0389 [M − C_6_H_10_O _5_− CH_3_ − OH + H]^+^, 97.0280	94.6		
**44**	(±)-Abscisic acid	17.55	C_15_H_20_O_4_	264.1359	−0.85	265.1484 [M + H]^+^	265.1481 [M + H]^+^	247.1332 [M − H_2_O + H]^+^, 229.1216 [M − 2H_2_O + H]^+^, 187.1108 [M − O − CH_2_ − COO + H]^+^	90.2		
**45**	*trans*-Cinnamic acid	18.16	C_9_H_8_O_2_	148.0518	−0.60	149.0232 [M + H]^+^	149.0231 [M + H]^+^	**144.9817**, **131.0493** [M − H_2_O + H]^+^, **121.0282**, 116.9669, ***107.0491***, ***105.0539*** [M − CO + H]^+^, 93.0698, 79.0545		Standard [28,30]	CR, CC, *C. cassia* leaves
**46**	4-Phenyl-3-buten-2-one	18.31	C_10_H_10_O	146.0730	−1.03	147.0803 [M + H]^+^	147.0803 [M + H]^+^	132.0567 [M − CH_3_ + H]^+^, 129.0699, 119.0854, 117.0698, 107.0489 [M − CH − CO + H]^+^, 91.0541 [M − C_3_H_4_O + H]^+^, 79.0542	95.3	[32,49]	*C. verum*
**47**	3-Tert-butyladipic acid	18.32	C_10_H_18_O_4_	202.1196	−1.36	201.1121 [M − H]^−^	201.1120 [M − H]^−^	183.1021 [M − OH − H]^−^, 156.8982 [M − COO − H]^−^, 139.1124	70.0	[30]	
**48**	*trans*-Cinnamaldehyde	18.84	C_9_H_8_O	132.0573	−0.19	133.0647 [M + H]^+^	133.0646 [M + H]^+^	**115.0540** [M − H_2_O + H]^+^, ***105.0697*** [M − CO + H]^+^, 103.0542, **91.0541** [M − CO − CH_2_ + H]^+^, **79.0542** [M − CO − C_2_H_2_ + H]^+^, **55.0178** [M − C_6_H_6_ + H]^+^	97.9	Standard, [30]	CR, CC
**49**	2-Methoxycinnamic acid	19.54	C_10_H_10_O3	178.0629	−0.51	161.0597 [M + H − H_2_O]^+^	161.0596 [M + H − H_2_O]^+^	**146.0366**, **133.1011** [M − H_2_O − CO + H]^+^, **119.0855** [M − CHO − OCH_3_ + H]^+^, ***105.0698*** [M − COOH − CH + H] ^+^, **91.0544** [M − CO − CH_2_ − OCH_3_ + H]^+^		Standard, [30]	CR, CC, *C. cassia* leaves
**50**	9*S*,13*R*-12-Oxophytodienoic acid	20.08	C_18_H_28_O_3_	292.2037	−0.68	293.2111 [M + H]^+^	293.2104 [M + H]^+^	275.2003 [M − H_2_O + H]^+^, 257.1893, 239.1799 [M − C_4_H_6_ + H]^+^, 229.1953, 163.1117, 159.1167, 147.1163 [M − C_7_H_14_ + H]^+^, 133.1012, 107.0855, 95.0853, 81.0698 [M − C_12_H_20_O_3_ + H]^+^, 69.0699	91.0	[50]	
**51**	Corchorifatty acid F	20.43	C_18_H_32_O_5_	328.2237	−0.10	327.2166 [M − H]^−^	327.2165 [M − H]^−^	309.2062, 291.1955, 242.9845 [M − C_5_H_4_ − OH − H]^−^, 239.1283, 229.1435, 221.1171, 211.1313, 185.1173, 183.1374, 171.101 [M − C_9_H_16_O_2_ − H]^−^, 137.0968, 97.0655, 85.0290 [M − C_13_H_22_O_4_ − H]^−^, 57.0343		[34,51]	
**52**	Deoxyphomalone	20.47	C_13_H_18_O_4_	238.1204	−0.29	239.1277 [M + H]^+^	239.1275 [M + H]^+^	221.1171, 205.1192 [M − 2OH + H]^+^, 179.0705 [M − C_2_H_5_ − OCH_3_ + H]^+^, 174.0678, 163.0750, 151.0753 [M − C_2_H_5_ − C_3_H_7_O + H]^+^, 137.0598 [M − 2OH − 2OCH_3_ − C_2_H_4_ − C_3_H_3_+H]^+^, 135.0799, ***107.0481***, 95.0861 [M − OH − 2OCH_3_ − C_2_H_5_ − C_4_H_3_O + H]^+^, 59.0490	74.4		
**53**	4-Ethylbenzaldehyde	20.73	C_9_H_10_O	134.0730	−0.14	135.0804 [M + H]^+^	135.0803 [M + H]^+^	120.0567, ***107.0490*** [M − CO + H]^+^, **105.0697** [M − C_2_H_6_ + H]^+^, 103.0542, 79.0542 [M − C_2_H_6_ − CO + H]^+^	92	[52]	CR
**54**	1-Naphthol	21.00	C_10_H_8_O	144.0573	−0.09	145.0648 [M + H]^+^	145.0647 [M + H]^+^	116.0575 [M − C − OH + H]^+^, 115.0541, 102.0468 [M − C_2_H_2_ − OH + H]^+^, 91.0539 [M − C_3_H_2_ − OH + H]^+^, 84.9598	89	[53,54]	CR
**55**	4-Methoxy cinnamaldehyde	21.02	C_10_H_10_O_2_	162.0679	0.04	163.0753 [M + H]^+^	163.0754 [M + H]^+^	145.0650, 135.0805 [M − CO + H]^+^, 133.0648, 110.0203 [M − C_3_H_3_O + H]^+^, 107.0491, ***105.0699*** [M − CO − OCH_3_ + H]^+^, 91.0542, 79.0542 [M − C_3_H_3_O − OCH_3_ + H]^+^, 55.0178	88.4	[46]	CR, CC
**56**	9,12,13-Trihydroxy-15-octadecenoic acid	21.72	C_18_H_34_O_5_	330.2393	−0.05	329.2322 [M − H]^−^	329.2322 [M − H]^−^	311.2227 [M − H_2_O −H]^−^, 293.2102 [M − 2H_2_O − H]^−^, 229.1433, 211.1331, 183.1383, 171.1018, 139.1123, 127.1120, 125.0975, 99.0812, 57.0342	90.0		
**57**	(−)-Caryophyllene oxide	22.32	C_15_H_24_O	220.1826	−0.53	221.1899 [M + H]^+^	221.1900 [M + H]^+^	203.1795, 175.1483 [M − O − 2CH_2_ −C + H]^+^, 161.1323 [M − 2CH_3_ − CO − CH + H]^+^, 147.1169 [M − 2CH_3_ − CO − CH − CH_2_ + H]^+^, 133.1010, 119.0855, 95.0855	92.9	[55]	CR, CC, *C. cassia* leaves
**58**	4-Methoxychalcone	28.61	C_16_H_14_O_2_	238.0992	0.66	239.1066 [M + H]^+^	239.1073 [M + H]^+^	221.0961, 193.1012, 178.0875, 161.0595 [M − C_6_H_6_ + H]^+^,133.0647 [M − C_7_H_6_O + H]^+^, 115.054, 105.0333 [M − C_6_H_6_ − C_2_H − OCH_3_ + H]^+^	86.7	[56]	*C. cassia*

^a^ indicated that the comprehensive score of molecular formula, molecular structure and MS^2^ fragment ions matching with the mzCloud database. ^b^ indicated that this compound has been isolated or identified from a certain plant. Bold is their MS^2^ fragment ion that matched the standard. Bold and italicize is the diagnostic ion for each compound.

Compound **48** obtained a precursor ionic peak at *m*/*z* 133.0645 in the ESI (+) mode; its molecular formula of the compound was C_9_H_8_O and it had a molecular weight of 132.0572 based on the elemental composition analysis. The matching fragments were mainly *m*/*z* 115.0540 [M − H_2_O + H]^+^, *m*/*z* 105.0696 [M − CO + H]^+^, *m*/*z* 91.0540 [M − CO − CH_2_ + H]^+^, *m*/*z* 79.0542 [M − CO − C_2_H_2_ + H]^+^ and *m*/*z* 55.0177 [M − C_6_H_6_ + H]^+^ (Table 1), which was consistent with the cleavage fragment of *trans*-cinnamaldehyde in the literature and database: the compound was, therefore, presumed to be *trans*-cinnamaldehyde [57]. The MS^2^ spectrum of compound **48** is shown in Appendix A, and the possible cleavage process of the positive ions was shown in Figure 2.

#### 2.2.2. Identification of Flavonoids

The RDA cleavage of the C-ring was the main cleavage of flavonoids, resulting in fragment ions and the loss of a series of neutral small molecules such as H_2_O, CO_2_, and CO by energy collisions under certain mass spectrometric conditions [25,58]. Flavonoid oxyglycosides generally lost sugar first, and then cleaved according to the cracking law of flavonoid skeleton structure [26,59]. Combined with the mass fragments information in the literature and standards, the ionic fragments at *m*/*z* 291/289, 303/301, 287/285 were selected, respectively, as the diagnostic ion for epicatechin-type, quercetin-type, and kaempferol-type [23]. The cracking of flavonoids and dihydroflavones was very similar, for example, epicatechin (**11**) and quercetin (**32**) demonstrated the cracking laws of epicatechin-type and quercetin-type, as shown in Figure 3 and Figure 4, the MS^2^ spectrum of epicatechin (**11**) is shown in Appendix A.

Compound **11** had the molecular formula of C_15_H_14_O_6_ which resulted from a precursor ion at *m*/*z* 291.0860 [M + H]^+^, and preeminently from an H_2_O loss at the fragment peak *m*/*z* 273.0768 [M − H_2_O + H]^+^ [26]. This was caused, first, by the loss of C_6_H_6_O_2_ obtaining *m*/*z* 179.0340 [M − C_6_H_6_O_2_ + H]^+^, then by the loss of CH_2_ and CO, obtaining *m*/*z* 165.0544 [M − C_6_H_6_O_2_ − CH_2_ + H]^+^ and *m*/*z* 153.0376 [M − C_6_H_6_O_2_ − CO + H]^+^. Moreover, losses of C_9_H_8_O_3_, C_8_H_9_O_4_, C_9_H_11_O_2_ and C_9_H_10_O_4_ formed *m*/*z* 125.0239 [M − C_9_H_8_O_3_ + H]^+^, *m*/*z* 119.0491 [M − C_8_H_9_O_4_ + H]^+^, *m*/*z* 139.0388 [M − C_9_H_11_O_2_ + H]^+^ and *m*/*z* 109.0290 [M − C_9_H_10_O_4_ + H]^+^ (Table 1, Figure 3), This was consistent with the cleavage fragment of epicatechin in the literature and standards; accordingly, the compound was presumed to be epicatechin [59].

Compound **32** had the molecular formula C_15_H_10_O_7_ which resulted from a precursor ion at *m*/*z* 303.0496 [M + H]^+^ in the ESI (+) mode; this was primarily caused by losses of H_2_O and CO and formed *m*/*z* 257.0446 [M − CO − H_2_O + H]^+^; 2CO loss then occurred and *m*/*z* 229.0491 [M − 2CO − H_2_O + H]^+^, and *m*/*z* 199.0434 [M − 3CO − H_2_O + H]^+^ were obtained. In contrast, a loss of C_7_H_4_O_2_ formed *m*/*z* 153.0183 [M − CO − C_7_H_6_O_2_ + H]^+^, followed by a loss of H_2_O to form *m*/*z* 133.0292 [M − CO − C_7_H_4_O_2_ − H_2_O + H]^+^ (Table 1, Figure 4). This was consistent with the cleavage fragment of quercetin in the literature [59] and standards, so the compound was presumed to be quercetin.

#### 2.2.3. Identification of Coumarins

The main cleavage of coumarins included the losses of OH, CH_3_ and CO_2_ from characteristic substituents and the continuous neutral loss of CO from the pyran ring [58,60]. The fragmentation rule for the coumarin glycosides was similar to that for coumarins, except for the initial losses of sugar moieties.

As an example, compound **31** obtained a quasi-ionic peak at *m*/*z* 147.0439 [M + H]^+^ in ESI (+) mode, and the molecular formula for this compound was C_9_H_6_O_2_. The matching fragments were mainly *m*/*z* 103.0541 [M + H − CO_2_]^+^ and *m*/*z* 91.0540 [M + H − 2CO]^+^ (Table 1). This was consistent with the cleavage fragment of coumarin in the literature [61,62] and standard, so **31** was presumed to be coumarin. The MS^2^ spectrum of compound **31** was shown in Appendix A, and the possible cleavage process of the positive ions is shown in Figure 5.

Identification information of other compounds were detailed in Appendix A.

### 2.3. Statistical Analysis

The MS data from different sources of CR and CC were imported into Compound Discoverer 3.2 qualitative analysis software for normalization and export, with a total of 1460 positive and negative ion fragments. The processed data were imported into SICMA 14.1 software for statistical analysis, resulting in the designation of 37 discrepant compounds. Analysis of variance was performed on the discrepant compounds using GraphPad Prism 7.0 software, resulting in 26 statistically significant (*p* < 0.05) discrepant compounds.

#### 2.3.1. Principal Component Analysis (PCA)

PCA analysis can clearly explain the correlation between a large number of variables and a small number of principal components with less data loss; this can effectively reduce the dimensionality of the multidimensional raw data. The results showed that the CR and CC samples could be clearly separated and classified into one category, indicating that there were differences in the chemical compositions of CR and CC. When statistical analysis was used, the QC samples should be included to evaluate the instrument stability. The results of the PCA analysis are shown in Appendix A.

#### 2.3.2. Orthogonal Partial Least Squares-Discriminant Analysis (OPLS-DA)

The difference in the chemical compositions of CR and CC was further studied using supervised OPLS-DA; the score diagram was shown in Appendix A. The results indicated that CR and CC can be clearly separated, which was consistent with the PCA results. The model fitting was also validated by setting the number of tests to 200. The R^2^ and Q^2^ points on the left of the data were lower than the original values on the right, and the regression line at the Q^2^ intersected the vertical axis below the original point with a negative nodal increment. The results of the model validation were R^2^Y = 0. 999 and Q^2^ = 0.952 (R^2^ in the validation results represented the matching of the model. The closer R^2^ to 1, the more complete the model was in describing the data; the closer Q^2^ to 1, the higher the predictive ability of the model) (Appendix A). Model evaluation results indicated that the model had robustness and no over-fitting. The commonly used variable importance for the project (VIP) was used to estimate for quality difference markers between groups. Variables with a VIP > 1 was generally considered to be meaningful to the model, the greater the VIP value was, the greater the contribution of the variable. Therefore, a VIP > 1 was selected to analyze the overall variability of CR and CC to find differential markers (Appendix A). A total of 627 characteristic peaks was determined for the differential components (VIP > 1) that contributed significantly to the classification of CR and CC (Appendix A). A total of 37 differential compounds with VIP > 1 was identified based on the analysis of compound retention times, accurate relative molecular masses, and cleavage information. Information on compounds with VIP > 1 were shown in Appendix A. The peak areas of these 37 compounds were subjected to *t*-tests using GraphPad Prism7.0 software, resulting in 26 chemical components with VIP > 1 and *p* < 0. 05, including compounds **1**–**3**, **5**, **8**, **16**, **17**, **19**, **24**, **26**, **27**, **31**–**36**, **38**, **40**–**42**, **45**, **46**, **48**, **53** and **58**, These comprised two organic acids, six phenolic acids, seven phenylpropanoids, eight flavonoids, two terpenoids and one coumarin. According to the peak areas and secondary fragment information, of the 26 significantly different compounds, 6 (**19**, **32**, **33**, **38**, **41** and **53**) were specific to CR, and 3 (**8**, **26**, and **36**) were specific to CC.

#### 2.3.3. Semi-Quantitative Analysis of CR and CC

A semi-quantitative analysis was further carried out to compare the intensity trends of CR and CC by calculating the relative peak areas of 26 compounds in 16 samples. The box-lots could visually observe the content changes between chemical compositions. The peak areas data for the 26 compounds in CR and CC were exported, and the boxplots produced by GraphPad Prism 7.0 software were further used to compare the relative content of the compounds in CR and CC, as shown in Figure 6.

The results (Figure 6) showed that the peak areas of compounds **1**, **16**, **19**, **24**, **27**, **31**, **32**, **33**, **34**, **38**, **40**, **41**, **42**, **45**, **46**, **53** and **58** in CR were higher than those in CC. Among these compounds, compounds **19**, **27**, **32**, **33**, **38**, **40** and **41** were flavonoids and flavonoid glycosides; **16**, **41**, **42**, **45**, **46**, **53** and **58** were phenylpropanoids; **1** and **24** were organic acids; **31** was coumarin and **34** was lignan glycoside. The peak areas of compounds **2**, **3**, **5**, **8**, **17**, **26**, **35**, **36** and **48** in CC were higher than those in CR. Among them, *trans*-cinnamaldehyde (**48**) was the major active components in both CR and CC. Compound **26** was flavonoid glycoside; **2**, **3**, **8** and **17** were organic acids; **35** and **36** were terpenoids and **5** was catechol.

Accordingly, the peak areas of flavonoids and flavonoid glycosides, and phenylpropanoids except *trans*-cinnamaldehyde in CR were higher than in CC, and the peak areas of terpenoids and organic acids in CC were higher than in CR.

### 2.4. Method Validation

The relative standard deviations (RSDs) for the precision, stability, and repeatability investigations into of coumarin, cinnamyl alcohol, cinnamic acid, dimethoxy cinnamic acid and cinnamaldehyde were all <5% (Table 2), indicating that the method had good precision, repeatability, and recovery. Additionally, a recovery range of 97−101% (RSD < 4%) indicated its high recovery and reliability (Appendix A).

### 2.5. Quantitative Determination of the Major Constituents in CR and CC Using HPLC

The established HPLC analysis method was subsequently used to determine the representative components in eight batches of CR and CC products. In the case of quantitative analysis, all the samples (eight batches of CR and CC) were extracted three times and analyzed by HPLC. The RSD value of the concentrations of these five standards were less than 5%. The HPLC chromatograms were shown in the Appendix A. The contents of the five compounds are summarized in Table 3, based on their respective calibration curves. The results showed that there were significant differences in the composition content of CR and CC. Among them, the content of *trans*-cinnamaldehyde in CC was about twice that in CR, and the content of *trans*-cinnamic acid in CC was similar to that in CR; the contents of coumarin, cinnamyl alcohol and 2-methoxycinnamic acid in CC were significantly higher than those in CC. We, therefore, believe that these five components play a key role in the different efficacies of CR and CC.

### 2.6. Cluster Analysis

The original data of our compound content was imported into Lianchuan Biotechnology’s advanced heat map statistics software, and the original data was normalized by Z-score to obtain the result (Appendix A). As our statistics result indicated, CR and CC were each clustered into one group. Five compounds were screened out and could be used as chemical markers for distinguishing CR and CC; Among the five, two markers including trans-cinnamaldehyde and cinnamyl alcohol (VIP value is greater than the other three compounds) made greater contributions to sample grouping than the other three ones.

### 2.7. Molecular Docking

The different screened constituents were docked to PGC1*α* (PDB ID: 1XB7), SIRT3 (PDB ID: 4BN4) and AMPK (PDB ID: 4CFF). CDOCKER_INTERACTION_ENERGY ≤ −5.0 kJ/mol was used to produce a better binding ability for the compounds and proteins. The results were shown in Appendix A and Figure 7.

The results showed that compounds **40**, **34**, **17**, **26** and **27** had the highest binding energy with Glycosylated Hemoglobin, Type A1C (HbA1c); compounds **1**, **32**, **16** and **19** had the highest binding energy with peroxisome proliferator-activated receptor-gamma coactivator-1alpha (PGC1*α*); compounds **24**, **2**, **45**, **42** and **3** had the highest binding energy with silent information regulator protein 3 (SIRT3); and compounds **40**, **33**, **34**, **27** and **32** had the highest binding energy with AMP-activated protein kinase (AMPK). Among these compounds, only **2**, **3**, **17, 26** had a higher content in CC. These results demonstrated that the special and high-concentration components in CR showed high docking scores of affinities with targets such as HbA1c and the proteins in the AMPK–PGC1–SIRT3 signaling pathway, suggesting the greater potentials for CR in treating DPN than for CC. The compounds with high protein binding energy were phenylpropanoid (**1**, **16**, **42**, **45**) and flavonoids (**19**, **26**, **27**, **40**).

## 3. Materials and Methods

### 3.1. Materials and Reagents

The following were used: the UPLC-Oribtrap-Exploris-120-MS liquid chromatography-mass spectrometry system (Thermo Scientific, Waltham, MA, USA); a KQ-500B CNC Ultrasonic Cleaner (Kunshan Ultrasonic Instrument Co., Ltd., Kunshan, China); methanol, acetonitrile, formic acid (chromatographic grade, Fisher, Waltham, MA, USA); ultra-pure water was freshly prepared using a Milli-Q system (Millipore, Milford, MA, USA); a high speed refrigerated centrifuge (Thermo Fisher, Karlsruhe, Germany); Watsons purified water. *trans*-Cinnamaldehyde (104-55-2), *trans*-cinnamic acid (621-82-9), cinnamyl alcohol (104-54-1), coumarin (91-64-5) and dimethoxy cinnamic acid (6099-03-2) were purchased from Chengdu purechem-standard co. LTD; epicatechin (490-46-0), taxifolin (480-18-3) and quercetin (522-12-3) were purchased from Shanghai yuanye Bio-Technology Co., Ltd.; and the purities of all the standards were greater than 98%. The eight batches of Cinnamomi ramulus (CR) and Cinnamomi cortex (CC) that were respectively the dried twigs and bark of *Cinnamomum cassia* Presl, were collected from different locations (Guangxi, Guangdong and Sichuan) at different harvest times (2019, 2020 and 2021), and stored in engineering technology research center for the comprehensive development and utilization of authentic medicinal materials in Henan province, as shown in Appendix A.

### 3.2. Preparation of Sample Solutions

Each CR and CC sample was weighed (1 g) accurately and extracted under reflux with 50 mL ultrapure water for 2 h. They were left to reach room temperature; water was then added to compensate for the weight loss of the extraction solution. A volume of 1 mL of the extraction solution was diluted with methanol to 2 mL; this was filtered through a 0.22 µm nylon filter membrane and centrifuged at 12,000 rpm for 15 min for UPLC-MS analysis. A volume of 1 mL of the extraction solution was directly filtered and centrifuged for HPLC quantitative analysis. Mixed standard solutions (*trans*-cinnamaldehyde, *trans*-cinnamic acid, epicatechin, 2-methoxycinnamic acid and quercitrin) were used as QC samples.

### 3.3. Preparation of Reference Solutions

An appropriate amount of the reference materials of coumarin, cinnamyl alcohol, cinnamic acid, 2-methoxycinnamic acid and cinnamaldehyde were measured and weighed precisely; methanol was added to dissolve them, and 1.0331, 1.8667, 0.6667, 1.1333 and 3.4 mg/L, respectively, were prepared for standby.

### 3.4. Chromatography and Mass Spectrometry Conditions

#### 3.4.1. UPLC Method for Qualitative Analysis

Qualitative analysis was performed on UPLC-Orbitrap-Exploris-120-MS and the preferred column was achieved on A HYPERSILGOLD Vanquish C18 (2.1 mm × 100 mm, 1.9 μm) for chromatographic separation. The mobile phase consisted of acetonitrile (A) and 0.1% aqueous formic acid (B), with a flow rate of 0.3 mL/min, and the gradient elution condition was set as follows: 0~4 min, 5~8% A; 4~10 min, 8~ 16% A; 10~15 min, 16% A; 15~22 min, 16~30%A; 22~25 min, 30% A; 25~32 min, 30~40% A; 32~35 min, 50% A; 35~40 min, 50~90% A. The injection volume was 1 µL and the column temperature was 25 °C.

#### 3.4.2. UPLC-MS Method for Qualitative Analysis

MS data was acquired in fast chromatography MS^2^ mode, the mass spectrometer parameters were set as follows: The ESI was used in negative ion mode (ESI−) and in positive ion mode (ESI+). The following settings were used: the spray voltage was 2.5 kV(−) and 3.5 kV(+); the UHPLC-MS/MS mode was applied with an Orbitrap resolution of 120,000 for full-MS and 15,000 for dd-MS^2^; the isolation window (*m*/*z*) was 2; the RF Lens% was 70; the sheath gas pressure was 45 Arb; the auxilliary gas pressure was 15 Arb; the sweep gas pressure was 0 Arb; the capillary temperature was 320 °C; vaporizer temperature was 350 °C; the scanning range was *m*/*z* 80~800; the stepped normalized collision energies (NCE) were 15, 30 and 45 eV.

#### 3.4.3. HPLC Method for Quantitative Analysis

The major active components including coumarin, cinnamyl alcohol, *trans*-cinnamic acid, 2-methoxycinnamic acid and *trans*-cinnamaldehyde were used for quantitative analysis.

Quantitative analysis was performed on a Waters HPLC E2695 and the preferred column was achieved on a Waters C18 (4.6 mm × 250 mm, 5 μm) for the chromatographic separation. The mobile phase was consisted of acetonitrile (A) and 0.1% aqueous formic acid (B), with a flow rate of 1 mL/min, and the gradient elution condition was set as follows: 0~13 min, 5~15% A; 13~20 min, 15~ 26% A; 20~25 min, 26% A; 25~28 min, 26~30%A; 28~38 min, 30%~40% A; 38~48 min, 40~60% A; 48~52 min, 60%~95% A; 52~57 min, 95% A. The injection volume was 20 µL, and column temperature was 30 °C.

To evaluate linearity, a series of standard solutions with appropriate concentrations were obtained by diluting standard compounds with methanol. The calibration curves were drawn with the quality of the reference substance as the abscissa (X) and the peak area as the ordinate (Y), they showed a good linear relationship (r ≥ 0.999) within the test ranges. The same sample solutions of CR and CC were continuously injected to verify the precision of the instrument. The same sample solutions were injected, separately, 0, 2, 4, 8, 12, and 24 h separately to check the stability of the test solution. Six sample solutions were prepared independently to verify the repeatability of the method. The accuracy of the method was evaluated by a recovery test. Recovery (%) = {[Found − (Original sample + Add)]/Add} ∗ 100.

#### 3.4.4. LC-MS Data Processing and Statistics

The UPLC-Orbitrap-Exploris-120-MS data were acquired by the Trancefinder 5.1 software; the UHPLC-MS/MS mode was applied with an Orbitrap resolution of 120,000 for full-MS and 15,000 for dd-MS^2^. The date was then analyzed using the Xcalibur3.0 software. Each raw data processed using Compound Discoverer 3.2 qualitative analysis software following a specific workflow (Appendix A). The screening steps of the target compound were: All ions presenting a signal over 5 times the background noise and a peak intensity over 10^5^ were taken into account to create the extracted ion chromatogram (EIC). MS and MS^2^ spectra were then used to identify ions by searching the mzCloud, mzValut, Chemspider and mass Lists databases. Then select the matching compounds with a tolerance of 5 ppm and more than 60% of the database matching degree as our preliminary identified compound. Finally, the compounds were reconfirmed by combined with the parent ion and the MS2 fragment ions extracted from their original data with those in the relevant literature. Additionally, the compounds were all rechecked by searching literature materials to exclude the non-natural products. The peak area of fragment ions was statistically analyzed to determine the statistically significant difference components, and then the marker compounds were clustered for the purpose of distinguishing CR and CC.

To evaluate relationships on the basis of similarities or differences between groups of multivariate data, multivariate analyses (PCA and OPLS-DA) were performed using SICMA 14.1. PCA results were displayed in the form of score plots. OPLS-DA was conducted using class information as the Y-variable; the results were shown in the form of score plots. The contribution of variables to the analysis was explained using variable importance in projection (VIP) scores. VIP scores are a weighted sum of squares of PLS weights, with scores larger than 1 indicating variables are important to the mode. T-tests were performed for compounds with VIP > 1: compounds with *p* value of <0.05 were considered significantly differentiated compounds.

#### 3.4.5. Molecular Docking

We used Discovery Studio to predict the docking of small molecule compounds with key proteins. The 3D structure of the compound constructed by ChemOffice software was saved in *mol2 format, and its energy was minimized. The 3D structure of the target protein was downloaded from the PDB data (https://www.rcsb.org/), accessed on 24 November 2022, and Discovery Studio 2020 software was used to perform operations such as water removal and hydrogenation on the protein and generate an effective single 3D conformation by minimizing the energy.

## 4. Conclusions

In this study, chemical compounds in CR and CC were analyzed and identified using UPLC-Oribtrap-Exploris-120-MS/MS: a total of 58 chemical components were identified. Unsupervised PCA and supervised OPLS-DA were used to assess the differences between CR and CC, and GraphPad Prism 7.0 was used to perform *t*-tests on the differential components. A total of 26 statistically significant differential compounds were obtained, in which the peak areas of flavonoids (**19**, **27**, **32**, **33**, **38**, **40** and **41**) and phenylpropanoids (**16**, **41, 42**, **45, 46**, **53** and **58**) except cinnamaldehyde in CR were higher than in CC, and the peak areas of terpenoids (**35** and **36**) and organic acids (**2**, **3**, **8** and **17**) in CC were higher than in CR. Additionally, HPLC was used to determine the concentrations and differentiating capacities of coumarin, cinnamyl alcohol, cinnamic acid, 2-methoxycinnamic acid and cinnamaldehyde, which were the major active ingredients in both CR and CC. The results showed that the content of cinnamaldehyde in CC was about twice that in CR, the content of cinnamic acid in CC was similar to that in CR, and the content of coumarin, cinnamyl alcohol and 2-methoxycinnamic acid in CR were significantly higher than that in CC. These five major components could be used as markers for successfully distinguishing CR and CC [63]. Compared to the reported method in the literature, the quantitative method was simpler, and demonstrated good stability and reproducibility. In addition, the different screened constituents were docked to HbA1c, PGC1*α*, SIRT3 and AMPK, and the results showed that the special and high-concentration components in CR showed high docking scores of affinities with targets such as HbA1c or proteins in the AMPK–PGC1–SIRT3 signaling pathway, suggesting the greater potentials of CR in treating DPN than of CC. Furthermore, the compounds with higher protein binding energy were phenylpropanoid (**1**, **16**, **42**, **45**) and flavonoids (**19**, **26**, **27**, **40**), from which it be inferred that flavonoids and phenylpropanoids might be an important material basis for the differential efficacies of CR and CC. The above results provide comparative information on the chemical profiles of CR and CC, as well as the groundwork for exploring the effective substances in each.

This is the first time that the compositional differences in the aqueous extracts of CR and CC by LC-MS have been analyzed, which directly reflects the material basis for their different functions under the usage of decoction in clinical practices. The high-content flavonoids and phenylpropanoids in CR may be the key material basis for dispersing wind and cold medicines, and terpenoids and organic acids may be the main active constituents for interior-warming medicines. In previous studies, flavonoids such as quercetin decreased blood glucose levels [64], phenylpropanoids inhibited platelet aggregation [65], and coumarin had suppressive effects on neuropathic cold allodynia in rats [66]. A further investigation of these CR differential substances may make it possible to find effective drugs for treating DPN. The previous studies were mostly focused on trans-cinnamaldehyde, while the CR and CC differential components would be easily acquired, combined with modern separation technology, and are worthy of further modern pharmacological research.

## Figures and Tables

**Figure 1 molecules-28-02015-f001:**
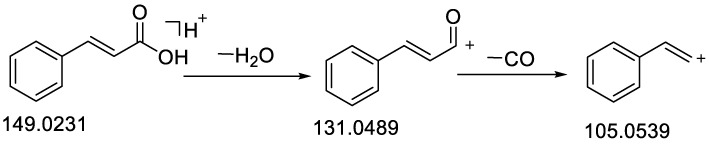
Fragmentation pathway of *trans*-cinnamic acid in positive ion mode.

**Figure 2 molecules-28-02015-f002:**
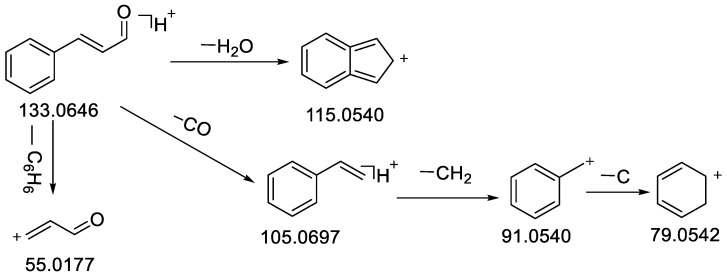
Fragmentation pathway of *trans*-cinnamaldehyde in positive ion mode.

**Figure 3 molecules-28-02015-f003:**
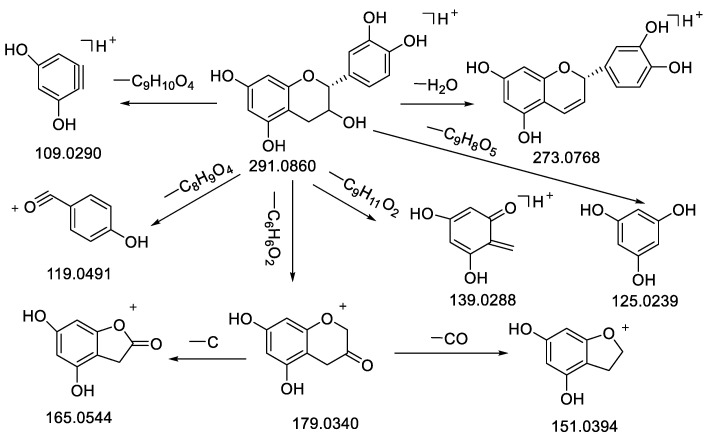
Fragmentation pathway of epicatechin in positive ion mode.

**Figure 4 molecules-28-02015-f004:**
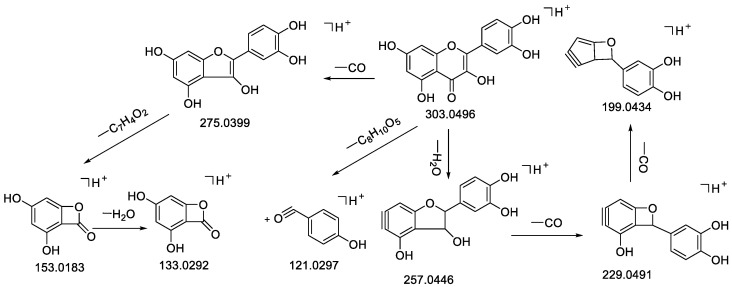
Fragmentation pathway of quercetin in positive ion mode.

**Figure 5 molecules-28-02015-f005:**
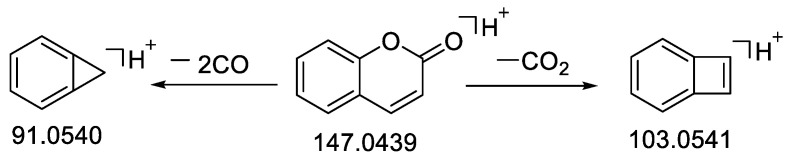
Fragmentation pathway of coumarin in positive ion mode.

**Figure 6 molecules-28-02015-f006:**
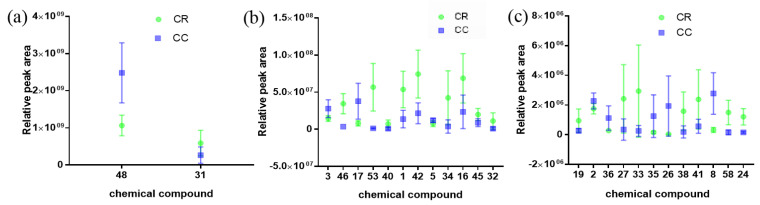
Box plots of the relative intensities of CR and CC: (**a**) compounds **48** and **31** with peak areas of 3.52 × 10^7^~3.17 × 10^9^; (**b**) compounds **3**, **46**, **17**, **53**, **40**, **1**, **42**, **5**, **34**, **16**, **45** and **32** with peak areas of 4.80 × 10^5^~1.48 × 10^8^; and (**c**) compounds **19**, **2**, **36**, **27**, **33**, **35**, **26**, **38**, **41**, **8**, **58** and **24** with peak areas of 3.15 × 10^4^~6.15 × 10^6^.

**Figure 7 molecules-28-02015-f007:**
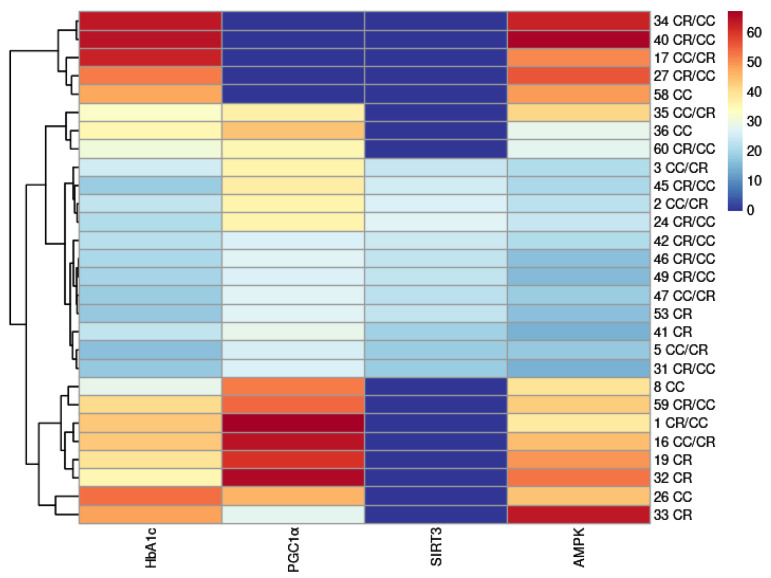
Heatmap of molecular docking.

**Table 2 molecules-28-02015-t002:** Linear-regression data, precision, stability, and repeatability of five standard compounds.

No.	Name	Linear	r	Linear Range (μg/mL)	Precision (RSD) n = 6	Stability (RSD) n = 6	Repeatability (RSD) n = 6	Average Recovery (%)	Recovery RSD (%)
**31**	Coumarin	Y = 1118.3X − 1.9591	0.9998	1.0331~1033.31	0.43%	0.83%	2.48%	100.09%	2.42%
**42**	Cinnamyl Alcohol	Y = 1377.2X − 0.0481	0.9997	0.0187~3.7300	2.30%	1.17%	2.81%	98.07%	2.77%
**45**	*trans*- Cinnamic acid	Y = 3219.7X − 1.3026	0.9999	0.6667~6.6670	0.47%	0.65%	1.60%	95.11%	2.09%
**48**	2-Methoxy cinnamic acid	Y = 12307X − 0.0527	0.9997	0.0133~2.6660	0.70%	0.74%	1.56%	98.30%	2.23%
**49**	*trans*- Cinnamaldehyde	Y = 449.01X − 5.4428	0.9995	108.30~2800	2.48%	3.43%	2.67%	97.36%	3.43%

**Table 3 molecules-28-02015-t003:** The contents of 5 analytes in 8 batches of CR and CC.

Sample	Coumarin (mg/g)	Cinnamyl Alcohol (mg/g)	*trans*-Cinnamic Acid (mg/g)	2-Methoxycinnamic Acid (mg/g)	*trans*-Cinnamaldehyde (mg/g)
CR-1	1.2038	0.0381	1.2205	0.0180	26.9798
CR-2	0.8544	0.0602	0.4867	0.0055	24.3722
CR-3	0.4209	0.1431	0.2674	0.0021	17.0832
CR-4	0.7392	0.0770	0.4862	0.0057	29.9873
CR-5	0.4531	0.0479	0.5113	0.0017	19.6756
CR-6	0.4834	0.0839	0.5245	0.0017	16.1460
CR-7	0.5107	0.0849	0.3960	0.0016	16.4895
CR-8	0.3563	0.1241	0.2769	0.0017	13.3894
CC-1	0.7534	0.0141	0.4164	0.0032	49.7582
CC-2	0.4307	0.0440	0.3474	0.0006	64.4378
CC-3	0.1880	0.0052	0.2370	0.0005	42.3951
CC-4	0.1611	0.0130	0.2914	0.0003	52.0785
CC-5	0.2030	0.0163	0.0811	0.0003	36.8182
CC-6	0.5814	0.0161	0.3301	0.0011	40.5779
CC-7	0.3655	0.0177	0.2932	0.0012	52.9302
CC-8	0.3851	0.0212	0.3169	0.0014	54.7701

## Data Availability

Data is contained within the article or Appendix A.

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
