# Peer review of "Identification of Differential Compositions of Aqueous Extracts of Cinnamomi Ramulus and Cinnamomi Cortex"

_molecules, 2023, doi:10.3390/molecules28052015_

Round 1

Reviewer 1 Report

Please refer to attachement.

Author Response

Dear reviewer 1,

Thank you very much for your letter and for the reviewers’ comments concerning our manuscript entitled “Identification of differential compositions of aqueous extracts of Cinnamomi ramulus and Cinnamomi cortex” (molecules-2186925). We have carefully revised our manuscript based on the editor’s and reviewers’ comments to improve its quality.

Our point-to-point responses to the comments were listed in the following pages. For your convenience, the changes were highlighted in red in the files named “Revised molecules-2186925” and “Revised Supplementary materials”. We hope our efforts will make this manuscript more acceptable for publication in molecules.

Thank you and all the reviewers again for the kind and helpful advice.

Best wishes.

Yours sincerely,

Li-ping Dai

Certificate of English Editing:

Reviewer 1

Comment 1. Table 1: The compound identification is doubtful. For example, compound 61 (norethandrolone) is a synthetic drug. How reliable is the compound identification step in this study?

Response: Many thanks for your comment. Indeed, compounds 58, 59 and 61 were all synthetic compounds through scientific literature search. We have to admit that it was due to our negligence, and these three compounds have been deleted in the text. To avoid this problem, all the compounds identified by matching the MS/MS data to the database have been rechecked by searching literature materials, to confirm them to be natural products. Additionally, the source of identified compounds (from CR or CC) was also added in Table 1 of the revised Manuscript.

So sorry to make you doubt the identification process of compounds. The compound identification step in this study was as follows: (1) Each raw data processed using Compound Discoverer 3.2 qualitative analysis software following a specific workflow (Figure S15). All ions presenting a signal over 5 times the background noise and a peak intensity over 105 were taken into account to create the extracted ion chromatogram (EIC). MS and MS2 spectra were then used to identify ions by searching the mzCloud, mzValut, Chemspider and Msss Lists databases. Then select the matching compounds with a tolerance of 5 ppm and more than 60% of the database matching degree as our preliminary identified compound. In order to ensure the accuracy of the results, the parent ion and the MS2 fragment ions of the compounds were extracted from the original data, and combined with those in the relevant literature to reconfirm the compound. Additionally, the compounds were all rechecked by searching literature materials to exclude the non-natural products. (2) The compound information of CR and CC (including molecular formula, molecular weight, MS1 and MS2 data and so on) was firstly established based on the previous literature. Then, search the possible precursor ion peaks in raw MS/MS data and compare its MS2 fragment ions with those in the literature to identify the compound. And the references have been provided in Table 1 of the revised Manuscript.

Figure S15. The workflow of compound screening.

Comment 2. Line 7-8: Please indicate in Table 1, the source of identification for each compound. Eg, if the compound was identified by matching the previous literature, the authors should provide the reference. If the compound was identified by matching to a library, indicate which library and the % of matching for the total score, RT, MW, or etc. For the 8 compounds identified by their authentic standard, please bold the MS2 of the compound that matched the standard.

Response: Many thanks for your comments. Actually, all the identified compounds have been analysed in the Manuscript or Supplementary material, which all have been provided the references. According to your suggestion, we have added the references in Table 1, and the total scores of the compounds identified by matching the data in database were also added in Table 1 of the revised Manuscript, as shown below.

For 8 compounds identified by their authentic standards, we have carefully rechecked and bolded their MS2 fragment ions that matched the standard in Table 1, as shown below.

Table 1. Identification of chemical constituents in CR and CC.

NO.

Name

RT

Formula

Calc. MW

Error

(ppm)

MS1 (m/z)

MS2 (m/z)

Total score (%) a

Ref.

Source b

1

Gentisic acid-5-O-glucoside

3.15

C13H16O9

316.0781

−4.28

315.0709 [M-H]

270.8695 [M−COOH+H] + 165.0183, 153.0185 [M−C6H10O5+H] +, 152.0108, 113.0240, 109.0293 [M−C6H10O5−COOH+H] +, 108.0211

91.2

[22]

2

Isovanillic acid

3.22

C8H8O4

168.0415

−2.06

167.0342 [M-H]

152.0116 [M−CH3−H] , 151.0226 [M−OH] , 123.0449 [M−COOH−H] , 108.0217 [M−COOH−CH−H] 

[23-25]

CR, C. cassia leaves

3

Gentisic acid

3.30

C7H6O4

154.0257

−4.16

153.0187 [M-H]

153.0187, 135.0182[M−H2O+H] +, 109.0288 [M−COOH+H] +, 85.0289, 81.0343 [M−COOH−CO+H] +, 68.9978

96.6

[26-27]

C. cassia 

4

Syringic acid

3.65

C9H10O5

198.0527

−0.55

199.0588 [M+H] +

181.0495[M−OH+H] + , 163.1478 [M−2H2O+H] +, 155.0166, 153.0764 [M−OH−OCH3+H] +, 95.0492

75.3

[26]

CR,CC,

C. cassia leaves

5

Catechol

3.80

C6H6O2

110.0363

−4.05

109.0291 [M-H]

108.0215 [M−2H] -, 93.7792 [M−OH−H] -, 81.6772

97.4

[28]

CR

6

Neochlorogenic acid

4.40

C16H18O9

354.0936

−4.21

353.0886 [M-H]

191.0550 [M−C9H6O3−H] , 179.0343[M−C7H10O5−H] , 161.0237 [M−C7H10O5−H2O−H] , 135.0445 [M−C7H10O5−CO2−H] , 111.0446

[29]

CR, CC

7

Salicylic acid

5.38

C7H6O3

138.0311

−0.47

137.0238[M-H]

119.0132 [M−H2O−H] , 108.8992 [M−CO−H]  , 93.0343 [M−COO−H]

62.9

[26], [30]

CR, CC

8

Citrinin

5.40

C13H14O5

250.0840

−0.46

251.0915 [M+H] +

233.0804 [M−H2O+H] +, 221.0807 [M−2CH3+H] +, 205.0857 [M−H2O−CO+H] +, 204.0785, 191.0701

75.5

9

4-Methoxy benzaldehyde

6.30

C8H8O2

136.0523

−0.97

137.0596 [M+H] +

122.0362 [M−CH3+H] +, 107.0490 [M−OCH2+H] +, 93.0590 [M−COCH3+H] +, 91.0543, 81.0698, 79.0543 [M−CO−OCH3+H] +

81.9

[23], [31]

CR, CC,

C. cassia leaves

10

Darendoside A

6.32

C19H28O11

432.1613

−4.37

431.1538 [M-H]

191.0547, 161.0446, 149.0447,113.0245, 99.0081, 89.0244

67.5

11

Epicatechin

8.41

C15H14O6

290.0788

−0.89

291.0860 [M+H] +

273.0768 [M−H2O+H] + , 249.0766, 179.0340 [M−C6H6O2+H] +, 165.0544 [M-C6H6O2−CH2+H] + , 151.0394, 139.0388 [M−C9H11O2+H] +,125.0239 [M−C9H8O3+H] +, 123.0439, 119.0491, 109.0290 [M−C9H10O4+H] +, 95.0490

99.4

Standard, [32]

CR

12

2,4,6-Trihydroxy-2-(4- hydroxybenzyl) -1-benzofuran-3(2H)-one

8.68

C15H12O6

288.0621

−4.34

287.0511 [M-H]

161.0233 [M−C6H6O3−H] , 131.2500, 125.0239 [M−C8H2O4−H]

88.7

CR, CC

13

Dihydrophaseic acid

8.75

C15H22O5

282.1454

−4.60

281.1382 [M-H]

237.1486 [M−CHO−H] , 201.1273 [M−CHO−H2O−OH−H] , 189.1278, 171.1171, 139.0758

76.8

CR

14

Catechin

8.83

C15H14O6

290.0777

−4.77

289.0703 [M-H]

289.0703 [M−H2O−H] , 271.0603, 245.08081, 203.07036, 151.0393, 125.0239, 109.0290

[30]

CC

15

Vanillin

9.28

C8H8O3

152.0472

−0.82

153.0545 [M+H] +

153.0545, 125.0595 [M−CO+H] +, 111.0441, 93.0333, 65.0387

90.7

[23], [26], [33]

C. cassia leaves

16

4-Acetyl-3-hydroxy-5-methylphenyl-β-D-glucopyranoside

10.08

C15H20O8

328.1144

−4.43

327.1071 [M-H]

165.0547 [M−C6H10O5−H] , 147.0446 [M−C6H10O5−H2O−H] , 121.0653 [M−C6H10O5−OH−CH3−CH2−OH] , 106.0416

91.7

[22]

17

Picconioside B

10.95

C26H40O12

544.2500

−3.66

543.2421 [M-H]

525.2305 [M−H2O−H] , 363.1800 [M−H2O−C6H10O5−H] , 381.1912, 167.1070, 165.0922, 101.0240, 89.0240, 59.0136

91.2

18

2 Methoxybenzoic acid

12.68

C8H8O3

152.0472

−0.92

153.0545 [M+H] +

153.0545, 135.0439, 111.0440, 105.0441, 95.0491 [M−CO2−CH2+H] +, 93.0699, 79.0541

96.3

[34], [35]

CR

19

Taxifolin

13.79

C15H12O7

304.0581

−0.67

305.0662 [M+H] +

287.0573 [M−H2O+H] +, 259.0591 [M−CO−H2O+H] +, 231.0652, 153.0188 [M−CO−C7H8O2+H] +, 149.0230

91.1

Standard,

[36], [37]

CR

20

Lyoniresinol- 3a-O

-β-D-glucopyranosid

13.89

C28H38O13

582.2289

−4.09

581.2209 [M-H]

566.1975, 535.1785, 419.1691 [M−C6H10O5−H] , 404.1459, 373.1275 [M−C6H10O5−3CH3−H] , 359.1110 [M−C6H10O5−4CH3−H] , 233.0812, 202.0624, 153.0549, 138.0316 [M−C6H10O5−OH−C14H18O6−H] , 101.0238

92.3

[28], [38], [39]

CC

21

4-Ethylphenol

14.12

C8H10O

122.0726

−4.32

121.0654 [M-H]

106.0419 [M−CH3−H] , 90.9232 [M−CH3−O−H] , 61.9880

87.9

[40], [41]

C. cassia

22

(−)-Lyoniresinol

14.19

C22H28O8

420.1766

−4.30

419.1694 [M-H]

373.1277 [M−3CH3−H] ,359.1119 [M−4CH3−H] , 313.0712, 221.0801, 180.0404, 139.0396, 134.0383

96.0

[28]

23

Lyoniside

14.33

C27H36O12

552.2186

−3.79

551.2105 [M-H]

536.1875, 419.1650, 389.1591, 374.1359, 373.1275 [M−C6H10O5−3CH3−H] , 359.1105 [M−C6H10O5−4CH3−H] , 341.1013, 325.1092, 233.0823, 119.0345, 113.0239

91.8

[28]

24

3-Oxoindane-1-carboxylic acid

14.41

C10H8O3

176.0472

−0.69

177.0545 [M+H] +

153.9367, 149.0596 [M−CO+H] +, 133.0646 [M−COO+H] +, 131.0490, 121.1010, 107.0490, 105.0693 [M−CO−COO+H]+, 93.0098 [M−CO−COO−C+H]+, 81.0700 [M−CO−COO−2C+H]+

72.2

25

3-Methoxy phenylacetic acid

14.52

C9H10O3

166.0622

−4.92

165.0549 [M-H]

147.0443, 136.9315 [M−OCH3−H]  , 121.0654, 106.0419, 96.9597 [M−C2H3O2−H]

[34]

26

2-(4-Hydroxyphenyl)-7-((3,4,5-trihydroxy-6-(hydroxymethyl) tetrahydro-2H-

3-pyran -2-yl) oxy) chroman-4-one

14.53

C21H22O9

418.1248

−3.75

417.1177 [M-H]

301.0338 [M−C6H10O5−H] , 255.0651, 153.0187 [M−C6H10O5−CO−C7H6O2−H] , 135.0082 [M−C6H10O5−CO−C7H6O2−H2O−H] , 119.0497, 91.0184

67.5

[36]

27

Quercetin-3β-D-glucoside

14.55

C21H20O12

464.0936

−4.03

463.0872 [M-H]

301.0338 [M−H−C6H10O5] , 300.0270, 271.0247

[29]

28

2- [1-(2H-1,3-Benzodiox ol- 5-yl) propan-2-yl]-6-metho xy-4-(prop-2-en-1-yl) phenol

14.74

C20H22O4

326.1516

−0.63

327.1593 [M+H] +

312.1348, 295.1328 [M−OCH2+H] +, 280.1095, 263.1071, 251.0001, 235.1122, 175.0758, 163.0753 [M−C10H12O2+H] +, 151.075, 137.0596, 133.0647, 103.0540, 98.9841

71.6

29

Cinnamylalcohol-6'-O-α-furanara-

binose-O-β-glucopyranoside

14.81

C20H28O10

428.1664

−4.38

427.1582 [M-H]

293.0861, 233.0650, 191.0549, 161.0451, 149.0448, 125.0239, 89.0240, 81.0344, 59.0136

85.9

[42]

CR

30

6-Methoxymellein

14.99

C11H12O4

208.0734

−0.55

209.0802 [M+H] +

191.0701, 181.0847, 177.0544, 163.0765, 149.0596 [M−COOCH+H] +, 131.0486, 121.0647 [M−2OH−OCH3−CO+H] +, 109.0647, 103.0540, 93.0698, 91.0540, 55.0177

85.8

.

31

Coumarin

15.18

C9H6O2

146.0366

−1.33

147.0439 [M+H] +

127.0543, 103.0541 [M−CO2+H] +, 91.0540 [M−2CO+H] +, 43.0242

96.8

Standard,

[42]

CR, CC, C. cassia leaves

32

Quercetin

15.36

C15H10O7

302.0424

−0.98

303.0496 [M+H] +

303.0496, 275.0399 [M−CO+H] +, 257.0446 [M−CO−H2O+H] +, 247.0590, 229.0491 [M−2CO−H2O+H] +, 199.0434 [M−3CO−H2O+H] +165.0178, 163.0389,

153.0183 [M−CO−C7H6O2+H] +, 133.0231[M−CO−C7H4O2−H2O+H] +, 121.0297, 111.0075

99.8

[24], [30]

CR

33

Quercitrin

15.36

C21H20O11

448.1003

−0.58

449.1087 [M+H] +

431.0983, 369.0594, 345.0606, 315.0494, 303.0497 [M+H−C6H9O4] +, 257.0439 [M−C6H9O4−CO−H2O+H] +, 229.0492 [M−C6H9O4−2CO−H2O+H] +, 129.0548, 85.0283, 71.0490

82.8

Standard

34

Graveobioside A

15.36

C26H28O15

580.1402

−4.46

579.1325 [M-H]

476.1080, 417.1531 [M−C6H10O5−H] , 300.0259 [M−C6H8O4−C5H10O4−H] , 271.0235 [M−C11H16O10−H] , 178.9979 [M−C17H20O11−H]

85.3

35

Libertellenone B

15.74

C20H26O4

330.1830

−0.32

331.1902 [M+H] +

313.1796 [M−H2O+H] +, 295.1676, 271.1686, 243.1763, 165.0911, 125.0565 [M−C12H14O3+H] +

75.2

36

Yucalexin P-17

16.06

C17H20O3

272.1412

−0.36

273.1488 [M+H] +

255.1391 [M−H2O+H] +, 245.1534, 227.1441 [M−H2O−CO+H] +, 203.1070, 149.0964 [M−CH3−C6H6O2+H] +, 82.8045 [M−C11H10O3+H] +

86.6

37

Azelaic acid

16.24

C9H16O4

188.1040

−4.55

187.0966 [M-H]

169.0861, 143.1072, 125.0966 [M−COOH−OH−H] , 123.0811, 97.0654 [M−2COOH−H] , 57.0343

95.4

[43], [44]

CR

38

Kaempferol

16.27

C15H10O6

286.0475

−0.89

287.0548 [M+H] +

287.0548, 258.0511 [M−CO+H] +, 183.0288, 165.0183, 153.0189 [M−CO−C7H6O+H] +, 133.0292,121.0281

99.2

[30]

CR, CC, C. cassia leaves

39

1-(Carboxymethyl) cyclohexane carboxylic acid

16.36

C9H14O4

186.0884

−4.58

185.0819 [M-H]

141.0916 [M−COO−H] , 104.0775 [M−C6H9−H]

87.2

40

Kaempferol-3-O-α-L-arabinopyranosyl-7-O-α-L-rhamnopyranoside

16.64

C26H28O14

564.1456

−4.12

563.1384 [M-H]

435.2045, 285.0416 [M−C6H8O4−C5H10O4−H] , 284.0316, 255.0286, 147.5166, 70.7867

[45]

41

2-Methoxy benzaldehyde

16.67

C8H8O2

136.0522

−1.38

137.0596 [M+H] +

109.0647, 107.0490 [M−OCH2+H] +, 93.0698 [M−CO−CH3+H] +, 81.0697, 79.0512 [M−CO−OCH3+H] +

90.9

[23]

CR, C. cassia leaves

42

Cinnamyl alcohol

17.22

C9H10O

134.0726

−0.89

117.0696 [M+H-H2O] +

117.0696 [M−H2O+H] +, 91.0540 [M−C2H4O+H] +, 78.2648 [M−C3H5O+H] +, 63.4672, 49.4958

standard

CR, CC, C. cassia leaves

43

4-Methylumbelliferyl-α-D-glucopyranoside

17.37

C16H18O8

338.1002

0.21

339.1075 [M+H] +

321.0970 [M−OH+H] +,177.0546 [M−C6H10O5+H] + ,145.0284 [M−C6H10O5−CH3−OH+H] +, 127.0389 [M−C6H10O5−CH3−OH+H] +, 97.0280

94.6

44

(±)-Abscisic acid

17.55

C15H20O4

264.1359

−0.85

265.1481 [M+H] +

247.1332 [M−H2O+H] +,, 229.1216 [M−2H2O+H] +,187.1108 [M−O−CH2−COO+H] +

90.2

45

trans-Cinnamic acid

18.16

C9H8O2

148.0518

−0.60

149.0231 [M+H] +

144.9817, 131.0493 [M−H2O+H] +, 121.0282, 116.9669, 107.0491, 105.0539 [M−CO+H] +, 93.0698, 79.0545

Standard

[23],[25]

CR, CC, C. cassia leaves

46

4-Phenyl-3-buten-2-one

18.31

C10H10O

146.0730

−1.03

147.0803 [M+H] +

132.0567 [M−CH3+H] +, 129.0699, 119.0854, 117.0698, 107.0489 [M−CH−CO+H] +, 91.0541 [M−C3H4O+H] +, 79.0542

95.3

[46], [47]

C. verum

47

3-Tert-butyladipic acid

18.32

C10H18O4

202.1196

−4.52

201.1120 [M-H]

183.1021 [M−OH−H] , 156.8982 [M−COO−H] , 139.1124

70.0

[25]

48

trans-Cinnamaldehyde

18.84

C9H8O

132.0573

−2.03

133.0646 [M+H] +

115.0540 [M−H2O+H] +, 105.0697 [M−CO+H] +, 103.0542, 91.0541 [M−CO−CH2+H] +, 79.0542 [M−CO−C2H2+H] +, 55.0178 [M−C6H6+H] +

97.9

Standard,

[25]

CR, CC

49

2-Methoxycinnamic acid

19.54

C10H10O3

178.0629

−0.51

161.0596 [M+H-H2O] +

146.0366,133.1011 [M−H2O−CO+H] +, 119.0855 [M−CHO−OCH3+H] +, 105.0698 [M−COOH−CH+H] +, 91.0544 [M−CO−CH2−OCH3+H] +

Standard,

[25]

CR, CC, C. cassia leaves

50

9S,13R-12-Oxophytodienoic acid

20.08

C18H28O3

292.2037

−0.68

293.2104 [M+H] +

275.2003 [M−H2O+H] +, 257.1893, 239.1799[M−C4H6+H] +, 229.1953, 163.1117, 159.1167, 147.1163[M−C7H14+H] +, 133.1012, 107.0855, 95.0853, 81.0698 [M−C12H20O3+H] +, 69.0699

91.0

[48]

51

Corchorifatty acid F

20.43

C18H32O5

328.2237

−3.80

327.2165 [M-H]

309.2062, 291.1955, 242.9845 [M−C5H4−OH−H] , 239.1283, 229.1435, 221.1171, 211.1313, 185.1173, 183.1374, 171.101 [M−C9H16O2−H] , 137.0968, 97.0655, 85.0290 [M−C13H22O4−H] , 57.0343

[29], [49]

52

Deoxyphomalone

20.47

C13H18O4

238.1204

−0.29

239.1275 [M+H] +

221.1171, 205.1192 [M−2OH+H] +, 179.0705 [M−C2H5−OCH3+H] +, 174.0678, 163.0750, 151.0753 [M−C2H5−C3H7O+H] +, 137.0598 [M−2OH−2OCH3−C2H4−C3H3+H] + , 135.0799, 107.0481, 95.0861 [M−OH−2OCH3−C2H5−C4H3O+H] +, 59.0490

74.4

53

4-Ethylbenzaldehyde

20.73

C9H10O

134.0730

−1.32

135.0803 [M+H] +

120.0567, 107.0490 [M−CO+H] +, 105.0697 [M−C2H6+H] +, 103.0542, 79.0542 [M−C2H6−CO+H] +

92

[50]

CR

54

1-Naphthol

21.00

C10H8O

144.0573

−1.19

145.0647 [M+H] +

116.0575 [M−C−OH+H] +, 115.0541, 102.0468 [M−C2H2−OH+H] +, 91.0539 [M−C3H2−OH+H] +, 84.9598

89

[51], [52]

CR

55

4-Methoxy cinnamaldehyde

21.02

C10H10O2

162.0679

−1.17

163.0754 [M+H] +

145.0650, 135.0805 [M−CO+H] +, 133.0648, 110.0203 [M−C3H3O+H] + , 107.0491, 105.0699 [M−CO−OCH3+H] +, 91.0542, 79.0542 [M−C3H3O− OCH3+H] +, 55.0178

88.4

[42]

CR, CC

56

9,12,13-Trihydroxy-15-octadecenoic acid

21.72

C18H34O5

330.2393

−3.94

329.2322 [M-H]

311.2227 [M−H2O−H] , 293.2102 [M−2H2O−H] , 229.1433, 211.1331, 183.1383, 171.1018, 139.1123, 127.1120, 125.0975, 99.0812, 57.0342

90.0

57

(−)-Caryophyllene oxide

22.32

C15H24O

220.1826

−0.53

221.1900 [M+H] +

203.1795, 175.1483 [M−O−2CH2−C+H] +, 161.1323 [M−2CH3−CO−CH+H] +, 147.1169 [M−2CH3−CO−CH−CH2+H] +, 133.1010, 119.0855, 95.0855

92.9

[53]

CR, CC, C. cassia leaves

58

4-Methoxychalcone

28.61

C16H14O2

238.0992

−0.66

239.1073 [M+H] +

221.0961, 193.1012, 178.0875, 161.0595 [M−C6H6+H] +,133.0647 [M−C7H6O+H] +, 115.054, 105.0333 [M−C6H6−C2H−OCH3+H] +

86.7

[54]

C. cassia

a indicated that the comprehensive score of molecular formula, molecular structure and MS2 fragment ions matching with the mzCloud database.

b indicated that this compound has been isolated or identified from a certain plant.

Comment 3. Table 1: As described in section 2.2.1, please bold all the diagnostic ions for each compound in Table 1.

Response: Thanks for your suggestion. We have bolded and italicized the diagnostic ions for each compound in Table 1 of the revised Manuscript.

Comment 4. Section 2.2.1: The description of trans-cinnamic acid is contradictory. The authors cited from previous literature m/z 105 is a diagnostic peak and shown in Figure 3. However, the peak is not shown in the spectrum of authentic standard and sample (Fig S3). Please explain.

Response: We apologize for our negligence and thanks for your comment. The diagnostic ion at m/z 105 was indeed not directly shown in the spectra of authentic standard and sample in the original Figure S3, which is because of a low relative abundance in MS2 spectra of both authentic standard and sample. Therefore, we have provided an enlarged local figure at m/z 105 in the revised Figure S5 of the revised Supplementary material, as shown below.

Figure S5. The MS2 spectrum of trans-cinnamic acid.

Comment 5. In addition, where are the MS2 peaks in Table 1 for compound 45 derived? Same query for all compounds.

Response: Thanks for your suggestion. So sorry for our negligence that some MS2 fragment ions (eg. compound 45) was not displayed completely in the original Table 1 because of low line-heights. We have carefully rechecked the MS2 data for all the compounds, and adjust the table size to make them display completely. Additionally, we also added the fragmentation process for the main MS2 ions of compounds in revised Table 1. And detailed analysis of all compounds was shown in Section 2.2 of Manuscript or Section 1-7 of Supplementary materials. 

Comment 6. Line 37: What is the meaning of “secondary mass spectrum”?

Response: We apologize for our negligence and thanks for your reminding. The “secondary mass spectrum” was a wrong expression, and has been modified as “the MS2 spectrum” (Line 140). And we have checked the manuscript carefully and corrected all the errors (Line 153, 169 and 200).

Comment 7. Line 41-45: Please indicate which data you are referring to when describing them. Eg, add (Table 1) at line 45. Same advice for the entire text.

Response: Thanks for your suggestion. These fragment ions were all referred to the data in Table 1. Therefore, we have added “(Table 1)” in the entire text when describing the relevant data (Line 138, 151, 181, 188 and 198).

Comment 8. Line 67: Figure S5 is referred to which result?

Response: Thanks for your reminding. We apologize for our negligence. Figure S5 should be referred to the MS2 spectrum of epicatechin (11), which now was Figure S7 (Line 169) in the revised manuscript. We have revised the sentence as “......take epicatechin (11) and quercetin (32) as examples......, as shown in Figure 3 and 4. And the MS2 spectra of epicatechin (11) was shown in Figure S7.”

Comment 9. Line 65: Compound 31 is different in Table 1.

Response: So sorry for the mistake. In fact, quercetin (31) in Line 167 should be quercetin (32), and we have corrected it.

Comment 10. Line 79: Where is Figure 8A?

Response: So sorry for the mistake. Figure 8A (Line 79) in the original manuscript is wrong, which should be Figure 4 in the revised version. We have revised it as “(Table 1, Figure 4)” in Line 188.

Comment 11. Line 86: duplication error

Response: Many thanks for your comment. We have revised the sentence (Line 192) as “The main cleavage of coumarins included the losses of OH, CH3 and CO2 from characteristic substituents, and the continuous neutral loss of CO from the pyran ring”.

Comment 12. Section 2.1: Please provide the data for the following claims:

  1. improved high sensitivity and good resolution by optimizing the mobile phase, flow rate,...and ion source parameters
  2. enhanced good resolution and better shapes by optimizing the column, mobile phase systems.
  3. reduced column pressure and analytical time by the said flow rate and column temperature. The claims were compared with what optimization?
  4. The signal response were MORE sensitive...MORE ion peak info...Data not shown. Please provide data for the claims.

Response: Thanks for your suggestions. In the manuscript, we detailed the optimization of the mobile phase, column type, column temperature and ion source parameters. Firstly, methanol/0.1% formic acid water and acetonitrile/0.1% formic acid water were investigated to achieve better separation. It was found that acetonitrile/0.1% formic acid water was the most suitable to obtain more peak and better peak shape. Secondly, we compared two types of columns (Waters Acquity UPLC BEH C18 and HYPERSILGOLD Vanquish C18) and found that the latter column had better resolution and more peaks. And different column temperatures (25 and 35 ℃) were tested and it had a good separation effect and higher peak intensity for the constituents at 35℃. In addition, we also compared the vaporizer temperature (320 and 350 â„ƒ) and the positive ion spray voltage (3.8 and 3.5 kV), and finally we found that the TIC high sensitivity and good resolution when the vaporizer temperature was 350℃ and the positive ion spray voltage was 3.5 kV. The above data were shown in Table S1. And the above information has been revised in Section 2.1 of the revised Manuscript.

Table S1. Optimization of the LC-MS conditions.

Comment 13. Line 99: Please cite the Figure no rather than page no.

Response: Thanks for your comment. The identification process of the compounds other than those in the text is detailed in Section 1-7 of the Supplementary materials and cited here (Line 205). According to your suggestion, “Page 21-28” have been changed to “Section 1-7”.

Comment 14. Figure 8: Please provide a PCA with QC samples. The QC samples should be included to evaluate the instrument stability when statistical analysis is used later.

Response: We apologize for our negligence and thanks for your reminding. We have provided the PCA with QC samples to evaluate the instrument stability in Figure S9 of the revised Supplementary materials. “In the qualitative analysis, in order to evaluate the stability of the instrument, mixed standards (QC) were repeated for 8 times. The RSDS of the retention time and intensity in QC samples were all less than 5%.” and “Mixed standard solutions (trans-cinnamaldehyde, trans-cinnamic acid, epicatechin, 2-methoxycinnamic acid and quercitrin) were used as QC samples.” were respectively added in Line 106 and 347 of the revised Manuscript.

Figure S9. 2D PCA score scatter plot for CR, CC and QC after eliminating outliers.

Comment 15. Line 127, 134 & 138: Incorrect Figure no

Response: So sorry for the mistakes. After carefully checking, it was found that the titles of “Figure S7” (Line 127) and “Figure S8” (Line 134) in the original Supplementary material were written reversely, and the title “Table S2. Information on CR and CC.” (Line 138) should be “Table S2. Compound information with VIP>1.”. And as the movement of figures in the text, “Figure S7”, “Figure S8” and “Table S2. Compound information with VIP>1.” have been revised as “Figure S11” (Line 233), “Figure S12” (Line 238) and “Table S2” (Line 243) in the revised Manuscript. And we have corrected the titles of Figure S11, Figure S12 and “Table S2” in the revised Supplementary material.

Comment 16. Line 136: Table S1 not found.

Response: Thanks for your reminding. So sorry for the wrong Figure number. Table S1 here should refer to “Compound information with VIP>1”. But in the original Supplementary material, there are Table S2, and “Table S2. Compound information with VIP>1.” should be Table S1. Accordingly, we have corrected it as “Table S2. Compound information with VIP>1.” in revised Supplementary material.

Comment 17. Line 138-145: Suggest to indicate in “Table S2” compounds that are significant in t-test, their fold change/trend (up/down) and which are specific in certain species only.

Response: Thanks for your suggestion. We have added the P value of t-test in Table S2. And we also added a separate column for Fold change. Fold change=log2 (ACR/ACC). A positive value indicated that the expression of compound in CR was higher than that in CC, while a negative value indicated that the expression of compound in CR was lower than that in CC. We have revised it as “Table S2. Compound information with VIP>1.”

Table S2. Compound information with VIP>1.

No.

Name

VIP

Fold change a

P-value b

3

Gentisic acid

1.60199

-0.93

0.0100

46

4-Phenyl-3-buten-2-one

1.55590

3.28

0.0001

55

4-Methoxycinnamaldehyde

1.51806

/

0.4002

24

3-Oxoindane-1-carboxylic acid

1.50622

2.93

0.0001

53

4-Ethylbenzaldehyde

1.48790

Specific in CR

0.0002

58

4-Methoxychalcone

1.44618

3.26

0.0005

8

Citrinin

1.44300

Specific in CC

0.0002

1

Gentisic acid 5-O-glucoside

1.39626

1.96

0.0010

48

trans-Cinnamaldehyde

1.38744

-1.22

0.0003

42

Cinnamyl alcohol

1.38222

1.88

0.0008

43

4-Methylumbelliferyl-α-D-glucopyranoside

1.34671

/

0.1463

40

Kaempferol 3-O-α-L-arabinopyranosyl -7-O-α-L-rhamnopyranoside

1.29033

7.41

0.0055

41

2-Methoxybenzaldehyde

1.23925

Specific in CR

0.0258

38

Kaempferol

1.23917

3.21

0.0116

34

Graveobioside A

1.22572

3.48

0.0114

5

Catechol

1.22299

-0.67

0.0231

16

4-Acetyl-3-hydroxy-5-methylphenyl β-D-glucopyranoside

1.21275

-1.55

0.0067

17

Picconioside B

1.21051

-2.24

0.0038

45

trans-Cinnamic acid

1.20494

-1.16

0.0065

26

2-(4-hydroxyphenyl)-7-((3,4,5-trihydroxy-6-(hydroxymethyl)tetrahydro-2H-pyran-2-yl)oxy)chroman-4-one

1.17656

Specific in CC

0.0191

32

Quercetin

1.16950

Specific in CR

0.0219

31

Coumarin

1.16629

1.15

0.0411

44

(±)-Abscisic acid

1.13767

/

0.0797

27

Quercetin-3β-D-glucoside

1.13751

2.81

0.0293

33

Quercitrin

1.13418

Specific in CR

0.0296

36

Yucalexin P-17

1.12735

Specific in CC

0.0126

7

Salicylic acid

1.12588

/

0.2768

49

2-Methoxycinnamic acid

1.11565

/

0.1049

2

Isovanillic acid

1.08152

-0.38

0.0374

22

(−)-Lyoniresinol

1.07830

/

0.1095

19

Taxifolin

1.07764

Specific in CR

0.0295

11

Epicatechin

1.05657

/

0.1232

6

Neochlorogenic acid

1.04831

/

0.0644

14

Catechin

1.02725

/

0.1559

35

Libertellenone B

1.00161

-0.98

0.0497

a Fold change=log2 (ACR/ACC). ACR and ACC represent the peak areas of the compounds in CR and CC. A positive value indicated that the expression of compound in CR was higher than that in CC, while a negative value indicated that the expression of compound in CR was lower than that in CC.

b P-value < 0.05 was significant.

Comment 18. Figure 10: Why the significant compounds are divided into 3 box plots?

Response: Thanks for your comment. Because there was the large difference in peak areas between compounds, their fold change/trend (up/down) for the most compounds, especially for the low-content compounds, cannot be clearly displayed in one box diagram. Thus, the significant compounds are divided into 3 box plots in Figure 6.

Comment 19. Line 179-180: Please describe how were the accuracy/recovery data calculated.

Response: Thanks for your suggestion. We have added the calculation method of recovery rate in the revised manuscript (Line 394). Recovery(%) ={[Found-(Original sample + Add)]/Add}*100. And the detailed experimental data of recovery rates was also added in Table S3 of the Supplementary material.

Table S3. Results of recovery rate

NO.

Name

Original

sample/ug

Add/ug

Found/ug

Recovery %

Mean %

RSD %

31

Coumarin

0.7071

73.5840

73.5840

73.5840

73.5840

73.5840

73.5840

1.4641

104.69%

100.09%

2.42%

0.7071

73.5840

73.5840

73.5840

73.5840

73.5840

73.5840

1.4173

98.21%

0.7071

73.5840

73.5840

73.5840

73.5840

73.5840

73.5840

1.4153

97.93%

0.7071

73.5840

73.5840

73.5840

73.5840

73.5840

73.5840

1.4358

100.77%

0.7071

73.5840

73.5840

73.5840

73.5840

73.5840

73.5840

1.4328

100.36%

42

Cinnamyl

alcohol

0.0366

0.0373

0.0724

95.96%

98.07%

2.77%

0.0366

0.0373

0.0728

97.13%

0.0366

0.0373

0.0718

94.50%

0.0366

0.0373

0.0743

101.02%

0.0366

0.0373

0.0734

98.59%

45

trans-

Cinnamic acid

0.8004

0.7867

1.5520

95.53%

95.11%

2.09%

0.8004

0.7867

1.5443

94.56%

0.8004

0.7867

1.5460

94.77%

0.8004

0.7867

1.5602

96.58%

0.8004

0.7867

1.5222

91.75%

49

2-Methoxy

cinnamic acid

0.0080

0.00798

0.0158

98.00%

98.30%

2.23%

0.0080

0.00798

0.0160

99.63%

0.0080

0.00798

0.0155

94.18%

0.0080

0.00798

0.0158

98.15%

0.0080

0.00798

0.0160

100.14%

48

trans-

Cinnamaldehyde

23.2100

22.8000

45.1864

96.39%

97.36%

3.43%

23.2100

22.8000

46.2753

101.16%

23.2100

22.8000

45.2955

96.87%

23.2100

22.8000

44.2427

92.25%

23.2100

22.8000

45.2077

96.48%

Figure 11: Please indicate the peak with a compound name in the mix standards' Spectrum.

Response: We have added the compound numbers of the peaks in the mix standards’ spectrum, and the Figure has been numbered Figure S13 and removed to the Supplementary material.

Figure S13. HPLC chromatograms for quantitative analyses. 31: Coumarin, 42: Cinnamyl alcohol, 45: trans-Cinnamic acid, 49: 2-Methoxycinnamic acid, 48: trans- Cinnamaldehyde 

Comment 20. Section 2.5: How did the authors ensure the each peak in CC and CR (Figure 11) was corresponded to a single compound rather than co-elution?

Response: Many thanks for the comment. In this experiment, the each peak had good resolution (Figure S13) in both mixed standard and samples, which was necessary for quantitative analyses by HPLC. Moreover, all the UV absorption spectra of the five compounds in CR and CC (Figures B and C) were pure and consistent with those in the mixed standard (Figure A). Accordingly, we determine that each peak for these five compounds were a single compound.

Figure S13. HPLC chromatograms for quantitative analyses. 31: Coumarin, 42: Cinnamyl alcohol, 45: trans-Cinnamic acid, 49: 2-Methoxycinnamic acid, 48: trans-Cinnamaldehyde 

Figure A. UV spectra of compounds in mixed standards.

Figure B. UV spectra of compounds in CR.

Figure C. UV spectra of compounds in CC.

Comment 21. Line 205: On what basis to claim the 5 compounds can be used as distinguishing markers for CR and CC, while Figure 12 indicates 0% contribution to the HCA?

Response: Thanks for your comments. These five compounds are the major active ingredients of CR and CC that can be displayed on their HPLC fingerprints (Figure S13) and their VIP value is greater than 1. In addition, the five compounds can be re-clustered to distinguish CR from CC, so the 5 compounds were used as distinguishing markers for CR and CC. But their contents in CR and CC show great differences, and the concentration of trans-cinnamaldehyde was indeed greatly higher than that of other four components (Table 3). Indeed, because of the distinct magnitudes of concentrations of detected markers, the original HETMAP could not obviously demonstrate the differentiating capability of each marker.

Table 3. The contents of 5 analytes in 8 batches of CR and CC.

Sample

Coumarin

(mg/g)

Cinnamyl alcohol

(mg/g)

trans-Cinnamic acid(mg/g)

2-Methoxycinnamic acid (mg/g)

trans-Cinnamaldehyde (mg/g)

CR-1

1.2038

0.0381

1.2205

0.0180

26.9798

CR-2

0.8544

0.0602

0.4867

0.0055

24.3722

CR-3

0.4209

0.1431

0.2674

0.0021

17.0832

CR-4

0.7392

0.0770

0.4862

0.0057

29.9873

CR-5

0.4531

0.0479

0.5113

0.0017

19.6756

CR-6

0.4834

0.0839

0.5245

0.0017

16.1460

CR-7

0.5107

0.0849

0.3960

0.0016

16.4895

CR-8

0.3563

0.1241

0.2769

0.0017

13.3894

CC-1

0.7534

0.0141

0.4164

0.0032

49.7582

CC-2

0.4307

0.0440

0.3474

0.0006

64.4378

CC-3

0.1880

0.0052

0.2370

0.0005

42.3951

CC-4

0.1611

0.0130

0.2914

0.0003

52.0785

CC-5

0.2030

0.0163

0.0811

0.0003

36.8182

CC-6

0.5814

0.0161

0.3301

0.0011

40.5779

CC-7

0.3655

0.0177

0.2932

0.0012

52.9302

CC-8

0.3851

0.0212

0.3169

0.0014

54.7701

Figure S14. Dendrograms of hierarchical cluster analysis (HCA) for CR and CC. (Original)

Accordingly, in our revised Manuscript, original data were firstly normalized and imported into software to regenerate the HETMAP figure. And then, the updated figure robustly indicated the clustering of samples as well as the contribution of each marker for sample clustering. Figure S14 in Supplementary material and the Cluster Analysis results in Section 2.6 of Manuscript have been updated.

Figure S14. Dendrograms of hierarchical cluster analysis (HCA) for CR and CC. (Updated)

Comment 22. Line 240-241: Sentence meaning is not matched the content of Table S2.

Response: Thanks for your reminding. So sorry for our negligence that there are two Table S2 in the original supplementary materials. “Table S2. Information on CR and CC.” should be right (Line 339), and in the revised Supplementary materials it has been modified as “Table S5. Information on CR and CC.”

Comment 23. Line 257: Suggest to add the info on column chemistry, i.e. C18, HILIC or etc.

Response: Thanks for your suggestion. The chromatographic columns used for the qualitative and quantitative experiments in this study were respectively a HYPERSILGOLD Vanquish C18 (2.1 mm×100 mm, 1.9 μm) and a Waters C18 (4.6 mm×250 mm, 5 μm), which have been added in Line 358 and 379 of the revised Manuscript.

Comment 24. Line 267: Please confirm the resolution of MS1. Why is it lower than MS2?

Response: So sorry for our negligence. The resolution of MS1 should be 12,0000, not 12,000. We have revised the sentence as “......was applied with resolution of the Orbitrap of 12,0000 for full-MS and 15000 for dd-MS2” (Line 397).

Comment 25. Line 275: Please confirm the column name.

Response: We apologize for our negligence and thanks for your reminding. The chromatographic column should be Waters C18 (4.6 mm×250 mm, 5 μm) (Line 379), and we have corrected it.

Comment 26. Line 281-284: What is the difference between using Tracefinder and Xcalibur for data acquisition?

Response: Thanks for your suggestion, and we apologize for our negligence. The statement in the manuscript was incorrect. In this study, Tracefinder was used to acquire data, and Xcalibur was used to analyze data. We have revised it as “... data were acquired by the Trancefinder 5.1 software, ...And the date was analyzed by using the Xcalibur3.0 software.” in revised Manuscript (Line 396-398).

Comment 27. Line 288: Please provide the info rather than using the phrase “so on”.

Response: Thanks for your suggestion. We have changed “so on” to “mass Lists” (Line 404).

Comment 28. Line 288: The steps for “targeted compound screening” is not provided in this section.

Response: Thanks for your suggestion. In the original Manuscript, we only briefly described the key steps “The principle of target compound screening”. Thus, in the revised manuscript, we detailed the steps of “targeted compound screening” in 3.4.4 as follows:

Each raw data processed using Compound Discoverer 3.2 qualitative analysis software following a specific workflow (Figure S15). The screening steps of the target compound were: All ions presenting a signal over 5 times the background noise and a peak intensity over 105 were taken into account to create the extracted ion chromatogram (EIC). MS and MS2 spectra were then used to identify ions by searching the mzCloud, mzValut, Chemspider and mass Lists databases. Then select the matching compounds with a tolerance of 5 ppm and more than 60% of the database matching degree as our preliminary identified compound. Finally, the compounds were reconfirmed by combined with the parent ion and the MS2 fragment ions extracted from their original data with those in the relevant literature. Additionally, the compounds were all rechecked by searching literature materials to exclude the non-natural products.

And the workflow of “targeted compound screening” by Compound Discoverer 3.2 was provided in Figure S15 of the revised Supplementary materials.

Figure S15. The workflow of compound screening with Compound Discoverer 3.2.

Comment 29. Table 4: For readability, please indicate the trend (up/down) of compounds in CR vs CC.

Response: Thanks for your suggestion. “Table 4” have been modified as “Table S4” in the revised Supplementary material. For readability, we also added a separate column for Fold change. Fold change=log2 (ACR/ACC). A positive value indicated that the expression of compound in CR was higher than that in CC, while a negative value indicated that the expression of compound in CR was lower than that in CC.

Table S4. Molecular docking fraction.

NO.

Fold changea

Name

-CDOCKER INTERACTION ENERGY/ kJ.mol-1

HbA1c

PGC1α

SIRT3

AMPK

1

1.96

Gentisic acid 5-O- glucoside

43.8038

67.0292

/

38.5316

2

-0.38

Isovanillic acid

23.2639

35.9152

26.4249

22.5498

3

-0.93

Gentisic acid

25.395

36.1031

23.8157

20.9892

5

-0.67

Catechol

16.6196

25.7563

18.3868

17.6189

8

Specific in CC

Citrinin

28.8092

52.3062

/

39.842

16

-1.55

4-Acetyl-3-hydroxy-5-methylphenyl β-D-glucopyranoside

43.7324

64.7307

/

45.1652

17

-2.24

Picconioside B

62.5016

/

/

50.7858

19

Specific in CR

Taxifolin

39.2912

60.4347

/

49.8092

24

2.93

Lyoniside

21.3959

36.4075

27.7012

23.9108

26

Specific in CC

2-(4-Hydroxyphenyl)-7-((3,4,5-trihydroxy-6-(hydroxymethyl) tetrahydro-2H- pyran-2-yl) oxy) chroman-4-one

53.7767

46.6389

/

44.2027

27

2.81

2-[1-(2H-1, 3-Benzod ioxol-5-yl)propan-2-yl]

52.3853

/

/

56.3881

31

1.15

Coumarin

17.7063

27.0471

18.5035

14.3744

32

Specific in CR

Quercetin

35.5995

66.1113

/

53.0311

33

Specific in CR

Quercitrin

48.2373

28.2954

/

63.6781

34

3.48

Graveobioside A

63.679

/

/

62.8879

35

-0.98

Libertellenone B

32.9261

36.9579

/

41.8519

36

Specific in CC

Yucalexin P-17

35.2668

44.4972

/

28.6512

40

7.41

Kaempferol 3-O-α-L-arabin opyranosyl-7-O-α-L-rhamnopyranoside

64.5903

/

/

66.4067

41

Specific in CR

2-Methoxy benzaldehyde

23.2034

28.5392

19.1136

14.6854

42

1.88

Cinnamyl alcohol

21.9571

26.9709

24.9043

21.6532

45

-1.16

trans-Cinnamic acid

18.5375

37.4825

25.3821

20.5352

46

3.28

4-Phenyl-3-buten-2-one

20.6405

27.6936

23.658

16.5148

48

-1.22

trans-Cinnamaldehyde

18.8605

27.4942

22.3644

18.7263

49

1.22

2-Methoxycinnamic acid

20.0461

26.8977

23.5866

15.9781

53

Specific in CR

4-Ethylbenzaldehyde

17.9064

27.6039

23.2434

16.3396

58

3.26

4-Methoxychalcone

30.8117

35.4958

/

27.9932

a Fold change=log2 (ACR/ACC). ACR and ACC represent the peak areas of the compounds in CR and CC. A positive value indicated that the expression of compound in CR was higher than that in CC, while a negative value indicated that the expression of compound in CR was lower than that in CC.

Comment 30. Line 317-218: This conclusion is confusing.

Response: Thanks for your suggestion. We have revised the sentence as “And 28 statistically significant differential compounds were obtained, in which the peak areas of flavonoids and flavonoid glycosides (19, 27, 32, 33, 38, 40 and 41), and phenylpropanoids (16, 41, 42, 45, 46, 53 and 58) except trans-cinnamaldehyde in CR were higher than CC, and the peak areas of terpenoids (35 and 36) and organic acids (2, 3, 8 and 17) in CC were higher than CR.”

Comment 31. Please engage a professional English proofreading service as many sentences are hanging or unclear.

Response: Thanks for your nice comment. Based on your suggestion, the manuscript has been edited with the aid of the Language Editing Service of MDPI to improve the English quality, and the certificate of English editing was shown as follow.

Reviewer 2 Report

Overall it was quite interesting study conducted by the authors yet there are some improvement need to be done. 

Please include the proper scientific botanical name of the plant studied especially in title or abstract or in plant material section.

Where is the plant material section in experimental part? Voucher specimen no? Plant material collection?

The LCMS/MS analysis data need to be detailed especially in table. Separate columns for relevant references for the metabolites detected previously from the same samples/species/genus. Add separate column for respective ionized molecular mass of the tentatively reported metabolites. I believe majority of the reported metabolites were identified tentatively through LCMS/MS analysis. 

I would suggest to add one more subsection on discussion to correlate all of the findings and discuss the findings in concise manner.

Author Response

Dear reviewer 2,

Thank you very much for your letter and for the reviewers’ comments concerning our manuscript entitled “Identification of differential compositions of aqueous extracts of Cinnamomi ramulus and Cinnamomi cortex” (molecules-2186925). We have carefully revised our manuscript based on the editor’s and reviewers’ comments to improve its quality.

Our point-to-point responses to the comments were listed in the following pages. For your convenience, the changes were highlighted in red in the files named “Revised molecules-2186925” and “Revised Supplementary materials”. We hope our efforts will make this manuscript more acceptable for publication in molecules.

Thank you and all the reviewers again for the kind and helpful advice.

Best wishes.

Yours sincerely,

Li-ping Dai

Reviewer 2

Comment 1. Please include the proper scientific botanical name of the plant studied especially in title or abstract or in plant material section.

Response: So sorry for the mistake. The proper scientific physical name “Cinnamomum cassia Presl” has been provided in the abstract. And the plants also were detailed as “The 8 batches of Cinnamomi ramulus (CR) and Cinnamomi cortex (CC) that were respectively the dried twigs and bark of Cinnamomum cassia Presl, were stored in the engineering technology research center for the comprehensive development and utilization of authentic medicinal materials in Henan province...” in the section of “3.1. Materials and Reagents”.

Comment 2. Where is the plant material section in experimental part? Voucher specimen no? Plant material collection?

Response: Many thanks for your nice comment. The plant materials have been detailed in 3.1 Materials and Reagents of revised manuscript. The 8 batches of CR and CC were stored in the key laboratory of traditional chinese medicine chemistry and resources of Henan province, and the sample sources, voucher specimen numbers and collection times were displayed in Table S5 of revised Supplementary material.

Table S5. Information on 8 batches of CR and CC.

No.

Origins

Sample sources

Collection times

Voucher specimen No.

CR-1

Guangxi

Kangmei pharmaceutical Bozhou Huatuo traditional Chinese medicine city

2020.07

20210910-001

CR-2

Guangxi

Henan Zhang Zhongjing Pharmacy Co., Ltd Zhengzhou City Store

2019.08

20210910-002

CR-3

Guangxi

Wuzhou city, guangxi

2021.08

20210910-003

CR-4

Guangxi

Kangmei pharmaceutical Bozhou Huatuo Traditional Chinese medicine city

2021.08

20210910-004

CR-5

Guangxi

Henan Zhang Zhongjing Pharmacy Co., Ltd Zhengzhou City Store

2019.08

20210910-005

CR-6

Guangdong

Zhaoqing, Guangdong

2021.09

20210910-006

CR-7

Guangdong

Zhongtang town, dongguan city, Guangdong

2021.06

20210910-007

CR-8

Sichuan

Kangmei pharmaceutical Bozhou Huatuo traditional Chinese medicine city

2021.08

20210910-008

CC-1

Guangxi

Kangmei pharmaceutical Bozhou Huatuo traditional Chinese medicine city

2021.08

20210911-001

CC-2

Guangxi

Kangmei pharmaceutical Bozhou Huatuo traditional Chinese medicine city

2022.08

20210911-002

CC-3

Guangxi

Guangxi

2021.09

20210911-003

CC-4

Guangdong

Dongguan, Guangdong province

2021.09

20210911-004

CC-5

Guangxi

Anhui bozhou economic development zone

2021.09

20210911-005

CC-6

Guangxi

Henan Zhang Zhongjing Pharmacy Co., Ltd Zhengzhou City Store

2019.08

20210911-006

CC-7

Guangxi

Kangmei pharmaceutical Bozhou Huatuo traditional Chinese medicine city

2021.08

20210911-007

CC-8

Sichuan

Kangmei pharmaceutical Bozhou Huatuo traditional Chinese medicine city

2021.08

20210911-008

Comment 3. The LCMS/MS analysis data need to be detailed especially in table. Separate columns for relevant references for the metabolites detected previously from the same samples/species/genus. Add separate column for respective ionized molecular mass of the tentatively reported metabolites. I believe majority of the reported metabolites were identified tentatively through LCMS/MS analysis.

Response: Thanks for your nice comment. After retrieving concerned literature, we separated columns for the source for the metabolites previously detected from the same samples/species/genus, as well as the relevant references in Table 1 (Ref. and source) of the revised manuscript. And we also added a separate column for the ionized molecular mass of compounds in Table 1 (MS1) of the revised manuscript. Indeed, the identification of compounds by LC-MS/MS analysis was limited.

Comment 4. I would suggest to add one more subsection on discussion to correlate all of the findings and discuss the findings in concise manner.

Response: Many thanks for your comment. We have added one more subsection on conclusion to correlate all of the findings and discuss the findings, as follows:

This is the first time to analyze the compositional differences in the aqueous extracts of CR and CC by LC-MS, which can directly reflect the material basis for their different functions under the usage of decoction in clinical. The high-content flavonoids and phenylpropanoids in CR maybe the key material basis for dispersing wind and cold medicines, and terpenoids and organic acids maybe the main active constituents of interior-warming medicines. In previous studies, flavonoids such as quercetin can decrease blood glucose levels, phenylpropanoids can inhibit platelet aggregation, and coumarin has the suppressive effects of neuropathic cold allodynia in rats. A further investigation of these CR substances is possible to find the effective drugs for treating DPN. The previous studies were mostly focused on trans-cinnamaldehyde, while the CR and CC differential components would be easily acquired combined with modern separation technology, and are worth of further modern pharmacological research.

Reviewer 3 Report

The manuscript submitted for evaluation contains a qualitative and quantitative comparison of the content of active compounds in aqueous extracts from the twigs and bark of Cinnamomum cassia. According to the information presented in the Introduction section, these raw materials are used in traditional Chinese medicine and are included in the Chinese Pharmacopoeia. However, they differ in use. Hence, as I understand it, the desire to identify the compounds responsible for the differences in action. Despite the fact that many elements in the work were correctly made and carefully described, the authors did not avoid serious faults. They are listed below.

1. How do patients receive cinnamon bark and twig preparations? Do they prepare them at home from dried raw material? Are they prepared in the pharmacy by extraction under reflux for 2 hours (i.e. as the samples were prepared in the study?). Or maybe there are drugs containing dried aqueous extracts. This problem requires clarification and justification as to why the samples in the study were prepared in this way and not otherwise.

2. I have not found any information on how the samples in each group (CR and CC) differ from each other in main text (only Table S2 in SM - it is misquoted in section 3.2.2.). Are there any significant limitations why there are only eight of them in each group? A larger number of samples would make it possible to separate the test set and evaluate the predictive capabilities of the obtained OPLS-DA model. The conducted internal validation does not allow for the assessment of the predictive capabilities of the model, but only for the assessment of its robustness and the occurrence of possible overfitting.

3. The section on methodology lacks information on the statistical analyzes carried out. Were the assumptions entitling to the use of the parametric t-test verified? In turn, the beginning of subsection 2.4. (part of the chapter results and discussion) should go to the methodology part.

4. Change the colors in Figure 10 to more contrasting ones.

5. In Table 2, it is worth adding designations (numbers) of compounds (previously used in figures and in the text).

6. In the case of qualitative analysis, was the analysis of the same sample repeated (e.g. three times)? Then it is worth adding the standard deviations of the obtained concentrations.

7. Do the obtained results - the indication of compounds whose concentrations differ significantly in water extracts in CR and CC - confirm the differences in the properties of these preparations. Or what other use of the results do the authors see?

Author Response

Dear reviewer 3,

Thank you very much for your letter and for the reviewers’ comments concerning our manuscript entitled “Identification of differential compositions of aqueous extracts of Cinnamomi ramulus and Cinnamomi cortex” (molecules-2186925). We have carefully revised our manuscript based on the editor’s and reviewers’ comments to improve its quality.

Our point-to-point responses to the comments were listed in the following pages. For your convenience, the changes were highlighted in red in the files named “Revised molecules-2186925” and “Revised Supplementary materials”. We hope our efforts will make this manuscript more acceptable for publication in molecules.

Thank you and all the reviewers again for the kind and helpful advice.

Best wishes.

Yours sincerely,

Li-ping Dai

Reviewer 3

Comment 1. How do patients receive cinnamon bark and twig preparations? Do they prepare them at home from dried raw material? Are they prepared in the pharmacy by extraction under reflux for 2 hours (i.e. as the samples were prepared in the study?). Or maybe there are drugs containing dried aqueous extracts. This problem requires clarification and justification as to why the samples in the study were prepared in this way and not otherwise.

Response: Thanks for your comments. Generally, cinnamon bark and twig that were CR and CC in the text, were made for Chinese herbal pieces, and dispensed into the Chinese medicine prescription and decocted in water under the doctor's advice. In the pharmacy or at home, the herbal pieces were soaked in cold water for 30 min and then decocted for 20-35 min twice.

In this experiment, the Chinese herbal pieces of CR and CC were refluxed with water for 2 hours, to obtain their aqueous extracts. The hot reflux extraction can obviously improve the quality of the soup, and very easy to operate, which is an ideal method for the preparation of soup in pharmacy. And many Chinese drugs preparations was produced under reflux. Therefore, we thought that the aqueous extracts of CR and CC obtained by reflux is matched with the traditional usage of decocting with water. 

Comment 2. I have not found any information on how the samples in each group (CR and CC) differ from each other in main text (only Table S2 in SM - it is misquoted in section 3.2.2.). Are there any significant limitations why there are only eight of them in each group? A larger number of samples would make it possible to separate the test set and evaluate the predictive capabilities of the obtained OPLS-DA model. The conducted internal validation does not allow for the assessment of the predictive capabilities of the model, but only for the assessment of its robustness and the occurrence of possible overfitting. 

Response: Thanks for your comments. So sorry for the misquotation of the original Table S2 in section 3.2.2., which have been corrected to be Table S5 in the revised Supplementary material. The eight batches of Cinnamomi ramulus (CR) and Cinnamomi cortex (CC) that were respectively the dried twigs and bark of Cinnamomum cassia Presl, were collected from different locations (Guangxi, Guangdong and Sichuan) at different harvest times (2019, 2020 and 2021), and stored in engineering technology research center for the compre-hensive development and utilization of authentic medicinal materials in Henan prov-ince, as shown in Table S5 (Line 334).

Table S5. Information on 8 batches of CR and CC.

No.

Origins

Sample sources

Collection times

Voucher specimen No.

CR-1

Guangxi

Kangmei pharmaceutical Bozhou Huatuo traditional Chinese medicine city

2020.07

20210910-001

CR-2

Guangxi

Henan Zhang Zhongjing Pharmacy Co., Ltd Zhengzhou City Store

2019.08

20210910-002

CR-3

Guangxi

Wuzhou city, guangxi

2021.08

20210910-003

CR-4

Guangxi

Kangmei pharmaceutical Bozhou Huatuo Traditional Chinese medicine city

2021.08

20210910-004

CR-5

Guangxi

Henan Zhang Zhongjing Pharmacy Co., Ltd Zhengzhou City Store

2019.08

20210910-005

CR-6

Guangdong

Zhaoqing, Guangdong

2021.09

20210910-006

CR-7

Guangdong

Zhongtang town, dongguan city, Guangdong

2021.06

20210910-007

CR-8

Sichuan

Kangmei pharmaceutical Bozhou Huatuo traditional Chinese medicine city

2021.08

20210910-008

CC-1

Guangxi

Kangmei pharmaceutical Bozhou Huatuo traditional Chinese medicine city

2021.08

20210911-001

CC-2

Guangxi

Kangmei pharmaceutical Bozhou Huatuo traditional Chinese medicine city

2022.08

20210911-002

CC-3

Guangxi

Guangxi

2021.09

20210911-003

CC-4

Guangdong

Dongguan, Guangdong province

2021.09

20210911-004

CC-5

Guangxi

Anhui bozhou economic development zone

2021.09

20210911-005

CC-6

Guangxi

Henan Zhang Zhongjing Pharmacy Co., Ltd Zhengzhou City Store

2019.08

20210911-006

CC-7

Guangxi

Kangmei pharmaceutical Bozhou Huatuo traditional Chinese medicine city

2021.08

20210911-007

CC-8

Sichuan

Kangmei pharmaceutical Bozhou Huatuo traditional Chinese medicine city

2021.08

20210911-008

During the epidemic, only 8 batches of samples were successfully collected and purchased, and we really should provide more batches of samples. However, many researchers also used 8 batches of samples for multivariate statistical analysis [1][2].

[1] Pei, H.T.; Su, W.Y.; Gui, M.; Dou, M.J.; Zhang, Y.X.; Wang, C.Z.; Lu, D. Comparative Analysis of Chemical Constituents in Different Parts of Lotus by UPLC and QToF-MS. Molecules (Basel, Switzerland). 2021; 26 (7): 1855.

[2] Wang, ZH.;, Zhao HX.; Tian, Lu.; Zhao, M.Y.; Xiao, Y,S.; Liu, Sh.Y.; Xiu, Y. Quantitative Analysis and Differential Evaluation of Radix Bupleuri Cultivated in Different Regions Based on HPLC-MS and GC-MS Combined with Multivariate Statistical Analysis. Molecules (Basel, Switzerland). 2022; 27(15): 4830.

I’m very sorry for our misunderstanding of internal validation. We have deleted the predictive ability of the model in the revised manuscript and modified it as “Model evaluation results indicated that the model had robustness and no over-fitting” (Line 233).

Comment 3. The section on methodology lacks information on the statistical analyzes carried out. Were the assumptions entitling to the use of the parametric t-test verified? In turn, the beginning of subsection 2.4. (part of the chapter results and discussion) should go to the methodology part.

Response: Thanks for your nice comments. We have added the methodology of multivariate statistical analysis (PCA and OPLS-DA) in section 3.4.4 of the revised manuscript. The peak areas of these 37 compounds (VIP>1) were subjected to t-tests using by GraphPad Prism7.0 software, and the results show that there were 26 chemical components with significant differences (P<0.05).

To evaluate relationships on the basis of similarities or differences between groups of multivariate data, multivariate analyses (PCA and OPLS-DA) were performed using SICMA 14.1. PCA results were displayed in the form of score plots. OPLS-DA was conducted using class information as the Y-variable; the results were shown in the form of score plots. The contribution of variables to the analysis was explained using variable importance in projection (VIP) scores. VIP scores are a weighted sum of squares of PLS weights, with scores larger than 1 indicating variables are important to the mode. T-tests were performed for compounds with VIP > 1: compounds with P value of <0.5 were considered significantly differentiated compounds.

We have moved the beginning of subsection 2.4. (part of the chapter results and discussion) “To evaluate linearity…… Recovery (%) = {[Found-(Original sample + Add)]/Add} *100”, to the methodology part, as shown in Section 3.4.3 of the revised Manuscript.

Comment 4. Change the colors in Figure 10 to more contrasting ones.

Response: Thanks for your suggestion. We have changed the colors of Figure 6 (Figure 10 in the original manuscript) to more contrasting ones, as shown below.

Figure 6. Box plots of the relative intensities of CR and CC.

Comment 5. In Table 2, it is worth adding designations (numbers) of compounds (previously used in figures and in the text).

Response: Thanks for your suggestion. We have added the designations (numbers) for the five compounds in Table 2, as shown below.

Table 2. Linear-regression data, precision, stability and repeatability of 5 standard compounds.

No.

Name

Linear

r

Linear Range

(μg/mL)

Precision

(RSD)

n=6

Stability

(RSD)

n=6

Repeatability

(RSD)n=6

Average

Recovery

Recovery

RSD

31

Coumarin

Y = 1118.3X - 1.9591

0.9998

1.0331~1033.31

0.43%

0.83%

2.48%

100.09%

2.42%

42

Cinnamyl

Alcohol

Y = 1377.2X - 0.0481

0.9997

0.0187~3.7300

2.30%

1.17%

2.81%

98.07%

2.77%

45

trans-

Cinnamic acid

Y = 3219.7X - 1.3026

0.9999

0.6667~6.6670

0.47%

0.65%

1.60%

95.11%

2.09%

48

2-Methoxy

cinnamic acid

Y = 12307X - 0.0527

0.9997

0.0133~2.6660

0.70%

0.74%

1.56%

98.30%

2.23%

49

trans-

Cinnamaldehyde

Y = 449.01X - 5.4428

0.9995

108.30~2800

2.48%

3.43%

2.67%

97.36%

3.43%

Comment 6. In the case of qualitative analysis, was the analysis of the same sample repeated (e.g. three times)? Then it is worth adding the standard deviations of the obtained concentrations.

Response: Thanks for your suggestion. In the case of qualitative analysis, only the mixed standards (QC) were repeated for 8 times. The RSDS of the retention time and intensity in QC samples were all less than 5%, as shown in the following Table A.

Table A. The RSDs of the retention time and intensity in the QC sample

Analyte

RSD (n=8)

Retention time

Intensity

Epicatechin

1.24%

3.52%

Quercitrin

0.97%

1.13%

trans-Cinnamic

0.88%

4.44%

trans-Cinnamaldehyde

0.65%

4.65%

2-Methoxycinnamic acid

2.23%

2.76%

In the case of quantitative analysis, all the samples (8 batches of CR and CC) were extracted for 3 times and analyzed by HPLC. The RSD value of the concentrations of these five standards were less than 5%, as shown in the following Table B.

Table B. The RSDs of the compound peak area in the same sample

NO.

RSD% (n=3)

Coumarin

Cinnamyl

trans-Cinnamic acid

2-Methoxycinnamic acid

trans-Cinnamaldehyde

CR-1

3.52

0.19

2.63

3.66

1.26

CR-2

4.47

1.23

3.02

3.48

1.37

CR-3

0.35

2.50

3.79

4.56

1.04

CR-4

2.40

4.76

4.71

4.23

0.66

CR-5

0.34

1.75

4.34

3.34

1.43

CR-6

4.86

3.05

4.09

2.81

1.32

CR-7

2.96

3.81

3.34

3.19

0.66

CR-8

0.06

3.44

4.32

3.81

1.44

CC-1

5.00

3.03

5.02

3.16

2.22

CC-2

0.61

4.30

4.43

4.88

4.26

CC-3

3.81

3.21

3.73

4.12

0.57

CC-4

2.05

3.01

4.29

2.40

2.45

CC-5

3.65

4.96

3.81

5.52

0.34

CC-6

1.52

2.77

4.58

4.02

1.66

CC-7

2.46

4.73

4.93

4.62

0.42

CC-8

1.33

1.47

4.22

3.86

1.61

Comment 7. Do the obtained results - the indication of compounds whose concentrations differ significantly in water extracts in CR and CC - confirm the differences in the properties of these preparations. Or what other use of the results do the authors see?

Response: Many thanks for your comment. This study firstly analysis the compositional differences in the aqueous extracts of CR and CC by LC-MS, which can directly reflect the material basis for their different functions under the usage of decoction in clinical. The results showed that the contents of flavonoids (19, 27, 32, 33, 38, 40 and 41) and phenylpropanoids (16, 41, 42, 45, 46, 53 and 58) except cinnamaldehyde in CR were higher than in CC, and these two types of compounds maybe the key material basis for dispersing wind and cold medicines. The contents of terpenoids (35 and 36) and organic acids (2, 3, 8 and 17) in CC were higher than in CR, and these two types of compounds maybe the main active constituents of interior-warming medicines. The previous studies were mostly focused on trans-cinnamaldehyde. This research provided the differential substances of CR and CC aqueous extracts, which can be further evaluated for the pharmacological activities, to clarify the material basis for the efficacy differences of CR and CC. A further investigation of these differential substances of CR is possible to find the effective drugs for treating DPN.

Round 2

Reviewer 1 Report

The quality of the manuscript has improved after the revision. Minor comments:

1.     Section 2.1: What were the samples (plant extract, standard or blank) used for the optimization? Please specify in the manuscript.

2.     Figure 6: Please explain in the figure legend why the compounds are divided into 3 plots.

Author Response

Dear reviewer 1,

Thank you very much for your letter and for the reviewers’ comments concerning our manuscript entitled “Identification of differential compositions of aqueous extracts of Cinnamomi ramulus and Cinnamomi cortex” (molecules-2186925). We have carefully revised our manuscript based on the editor’s and reviewers’ comments to improve its quality.

The aqueous extract of CR was used for optimization of the LC-MS conditions, and we have specified in Line 90 of the revised manuscript. We have added a legend for Figure 6 to explain why the compounds are divided into 3 plots. For your convenience, the changes were highlighted in blue in the files named “Revised-1 molecules-2186925”. We hope our efforts will make this manuscript more acceptable for publication in molecules.

Thank you and all the reviewers again for the kind and helpful advice.

Best wishes.

Yours sincerely,

Li-ping Dai

Reviewer 1

Comment 1. Section 2.1: What were the samples (plant extract, standard or blank) used for the optimization? Please specify in the manuscript.

Response: Many thanks for your comment. The aqueous extract of CR was used for optimization of the LC-MS conditions, and We have added “for aqueous extract of CR” in Line 90 of the revised manuscript.

Comment 2. Figure 6: Please explain in the figure legend why the compounds are divided into 3 plots.

Response: Many thanks for your nice comment. For Figure 6, because there was the large difference in peak areas between compounds, their fold change/trend (up/down) for the most compounds, especially for the low-content compounds, cannot be clearly displayed in one box diagram. Thus, the significant compounds are divided into 3 box plots in Figure 6.

According to your suggestion, we have added a legend for Figure 6 in the revised manuscript as “(a) compounds 48 and 31 with peak areas of 3.52×107~3.17×109; (b) compounds 3, 46, 17, 53, 40, 1, 42, 5, 34, 16, 45 and 32 with peak areas of 4.80×105~1.48×108; (c) compounds 19, 2, 36, 27, 33, 35, 26, 38, 41, 8, 58 and 24 with peak areas of 3.15×104~6.15×106.”. 

Figure 6. Box plots of the relative intensities of CR and CC. (a) compounds 48 and 31 with peak areas of 3.52×107~3.17×109; (b) compounds 3, 46, 17, 53, 40, 1, 42, 5, 34, 16, 45 and 32 with peak areas of 4.80×105~1.48×108; (c) compounds 19, 2, 36, 27, 33, 35, 26, 38, 41, 8, 58 and 24 with peak areas of 3.15×104~6.15×106.

Reviewer 2 Report

Overall, thanks for amending the suggested corrections. One more correction that I would like to suggest is that the calculated theoretical mass in Table 1 of respective compounds should be represented with an ionized form including the charge, respectively. 

Author Response

Dear reviewer 2,

Thank you very much for your letter and for the reviewers’ comments concerning our manuscript entitled “Identification of differential compositions of aqueous extracts of Cinnamomi ramulus and Cinnamomi cortex” (molecules-2186925). We have carefully revised our manuscript based on the editor’s and reviewers’ comments to improve its quality.

According to your suggestion, We also separated a column for theoretical mass of respective compounds in Table 1 of the revised manuscript. Our responses to the comment were listed in the following pages. For your convenience, the changes were highlighted in blue in the files named “Revised-1 molecules-2186925”. We hope our efforts will make this manuscript more acceptable for publication in molecules.

Thank you and all the reviewers again for the kind and helpful advice.

Best wishes.

Yours sincerely,

Li-ping Dai

Reviewer 2

Comment 1. One more correction that I would like to suggest is that the calculated theoretical mass in Table 1 of respective compounds should be represented with an ionized form including the charge, respectively.

Response: Many thanks for your comment. According to your suggestion, we have added a separate column for theoretical mass of respective compounds in Table 1. The deviation value (ppm) was also carefully checked and revised, with the theoretical mass in 4 decimal places.

Table 1. Identification of chemical constituents in CR and CC.

NO.

Name

RT

Formula

Calc. MW

Error

(ppm)

Theoretical mass (m/z)

Experimental mass (m/z)

MS2 (m/z)

Total score (%) a

Ref.

Source b

1

Gentisic acid-5-O-glucoside

3.15

C13H16O9

316.0781

−0.50

315.0711 [M-H]

315.0709 [M-H]

270.8695 [M−COOH+H] + 165.0183, 153.0185 [M−C6H10O5+H] +, 152.0108, 113.0240, 109.0293 [M−C6H10O5−COOH+H] +, 108.0211

91.2

[22]

2

Isovanillic acid

3.22

C8H8O4

168.0415

1.88

167.0339 [M-H]

167.0342 [M-H]

152.0116 [M−CH3−H] , 151.0226 [M−OH] , 123.0449 [M−COOH−H] , 108.0217 [M−COOH−CH−H] 

[23-25]

CR, C. cassia leaves

3

Gentisic acid

3.30

C7H6O4

154.0257

3.04

153.0182 [M-H]

153.0187 [M-H]

153.0187, 135.0182[M−H2O+H] +, 109.0288 [M−COOH+H] +, 85.0289, 81.0343 [M−COOH−CO+H] +, 68.9978

96.6

[26-27]

C. cassia 

4

Syringic acid

3.65

C9H10O5

198.0527

−4.53

199.0601 [M+H] +

199.0588 [M+H] +

181.0495[M−OH+H] + , 163.1478 [M−2H2O+H] +, 155.0166, 153.0764 [M−OH−OCH3+H] +, 95.0492

75.3

[26]

CR,CC,

C. cassia leaves

5

Catechol

3.80

C6H6O2

110.0363

4.05

109.0284 [M-H]

109.0291 [M-H]

108.0215 [M−2H] -, 93.7792 [M−OH−H] -, 81.6772

97.4

[28]

CR

6

Neochlorogenic acid

4.40

C16H18O9

354.0936

−4.21

353.0892 [M-H]

353.0886 [M-H]

191.0550 [M−C9H6O3−H] , 179.0343[M−C7H10O5−H] , 161.0237 [M−C7H10O5−H2O−H] , 135.0445 [M−C7H10O5−CO2−H] , 111.0446

[29]

CR, CC

7

Salicylic acid

5.38

C7H6O3

138.0311

3,50

137.0233[M-H]

137.0238[M-H]

119.0132 [M−H2O−H] , 108.8992 [M−CO−H]  , 93.0343 [M−COO−H]

62.9

[26], [30]

CR, CC

8

Citrinin

5.40

C13H14O5

250.0840

−0.46

251.0916 [M+H] +

251.0915 [M+H] +

233.0804 [M−H2O+H] +, 221.0807 [M−2CH3+H] +, 205.0857 [M−H2O−CO+H] +, 204.0785, 191.0701

75.5

9

4-Methoxy benzaldehyde

6.30

C8H8O2

136.0523

−0.77

137.0597 [M+H] +

137.0596 [M+H] +

122.0362 [M−CH3+H] +, 107.0490 [M−OCH2+H] +, 93.0590 [M−COCH3+H] +, 91.0543, 81.0698, 79.0543 [M−CO−OCH3+H] +

81.9

[23], [31]

CR, CC,

C. cassia leaves

10

Darendoside A

6.32

C19H28O11

432.1613

−2.29

431.1547 [M-H]

431.1538 [M-H]

191.0547, 161.0446, 149.0447,113.0245, 99.0081, 89.0244

67.5

11

Epicatechin

8.41

C15H14O6

290.0788

−1.08

291.0863 [M+H] +

291.0860 [M+H] +

273.0768 [M−H2O+H] + , 249.0766, 179.0340 [M−C6H6O2+H] +, 165.0544 [M-C6H6O2−CH2+H] + , 151.0394, 139.0388 [M−C9H11O2+H] +,125.0239 [M−C9H8O3+H] +, 123.0439, 119.0491, 109.0290 [M−C9H10O4+H] +, 95.0490

99.4

Standard, [32]

CR

12

2,4,6-Trihydroxy-2-(4- hydroxybenzyl) -1-benzofuran-3(2H)-one

8.68

C15H12O6

288.0621

−4.34

287.0521 [M-H]

287.0511 [M-H]

161.0233 [M−C6H6O3−H] , 131.2500, 125.0239 [M−C8H2O4−H]

88.7

CR, CC

13

Dihydrophaseic acid

8.75

C15H22O5

282.1454

−0.53

281.1383 [M-H]

281.1382 [M-H]

237.1486 [M−CHO−H] , 201.1273 [M−CHO−H2O−OH−H] , 189.1278, 171.1171, 139.0758

76.8

CR

14

Catechin

8.83

C15H14O6

290.0777

−1.26

289.0707 [M-H]

289.0703 [M-H]

289.0703 [M−H2O−H] , 271.0603, 245.08081, 203.07036, 151.0393, 125.0239, 109.0290

[30]

CC

15

Vanillin

9.28

C8H8O3

152.0472

−0.82

153.0548 [M+H] +

153.0545 [M+H] +

153.0545, 125.0595 [M−CO+H] +, 111.0441, 93.0333, 65.0387

90.7

[23], [26], [33]

C. cassia leaves

16

4-Acetyl-3-hydroxy-5-methylphenyl-β-D-glucopyranoside

10.08

C15H20O8

328.1144

−1.05

327.1074 [M-H]

327.1071 [M-H]

165.0547 [M−C6H10O5−H] , 147.0446 [M−C6H10O5−H2O−H] , 121.0653 [M−C6H10O5−OH−CH3−CH2−OH] , 106.0416

91.7

[22]

17

Picconioside B

10.95

C26H40O12

544.2500

−3.66

543.2438 [M-H]

543.2421 [M-H]

525.2305 [M−H2O−H] , 363.1800 [M−H2O−C6H10O5−H] , 381.1912, 167.1070, 165.0922, 101.0240, 89.0240, 59.0136

91.2

18

2 Methoxybenzoic acid

12.68

C8H8O3

152.0472

−0.92

153.0546 [M+H] +

153.0545 [M+H] +

153.0545, 135.0439, 111.0440, 105.0441, 95.0491 [M−CO2−CH2+H] +, 93.0699, 79.0541

96.3

[34], [35]

CR

19

Taxifolin

13.79

C15H12O7

304.0581

2.03

305.0656 [M+H] +

305.0662 [M+H] +

287.0573 [M−H2O+H] +, 259.0591 [M−CO−H2O+H] +, 231.0652, 153.0188 [M−CO−C7H8O2+H] +, 149.0230

91.1

Standard,

[36], [37]

CR

20

Lyoniresinol- 3a-O

-β-D-glucopyranosid

13.89

C28H38O13

582.2289

−4.09

581.2230 [M-H]

581.2209 [M-H]

566.1975, 535.1785, 419.1691 [M−C6H10O5−H] , 404.1459, 373.1275 [M−C6H10O5−3CH3−H] , 359.1110 [M−C6H10O5−4CH3−H] , 233.0812, 202.0624, 153.0549, 138.0316 [M−C6H10O5−OH−C14H18O6−H] , 101.0238

92.3

[28], [38], [39]

CC

21

4-Ethylphenol

14.12

C8H10O

122.0726

4.32

121.0649 [M-H]

121.0654 [M-H]

106.0419 [M−CH3−H] , 90.9232 [M−CH3−O−H] , 61.9880

87.9

[40], [41]

C. cassia

22

(−)-Lyoniresinol

14.19

C22H28O8

420.1766

−1.53

419.1700 [M-H]

419.1694 [M-H]

373.1277 [M−3CH3−H] ,359.1119 [M−4CH3−H] , 313.0712, 221.0801, 180.0404, 139.0396, 134.0383

96.0

[28]

23

Lyoniside

14.33

C27H36O12

552.2186

−3.79

551.2123 [M-H]

551.2105 [M-H]

536.1875, 419.1650, 389.1591, 374.1359, 373.1275 [M−C6H10O5−3CH3−H] , 359.1105 [M−C6H10O5−4CH3−H] , 341.1013, 325.1092, 233.0823, 119.0345, 113.0239

91.8

[28]

24

3-Oxoindane-1-carboxylic acid

14.41

C10H8O3

176.0472

−0.69

177.0546 [M+H] +

177.0545 [M+H] +

153.9367, 149.0596 [M−CO+H] +, 133.0646 [M−COO+H] +, 131.0490, 121.1010, 107.0490, 105.0693 [M−CO−COO+H]+, 93.0098 [M−CO−COO−C+H]+, 81.0700 [M−CO−COO−2C+H]+

72.2

25

3-Methoxy phenylacetic acid

14.52

C9H10O3

166.0622

1.69

165.0546 [M-H]

165.0549 [M-H]

147.0443, 136.9315 [M−OCH3−H]  , 121.0654, 106.0419, 96.9597 [M−C2H3O2−H]

[34]

26

2-(4-Hydroxyphenyl)-7-((3,4,5-trihydroxy-6-(hydroxymethyl) tetrahydro-2H-

3-pyran -2-yl) oxy) chroman-4-one

14.53

C21H22O9

418.1248

−0.74

417.1180 [M-H]

417.1177 [M-H]

301.0338 [M−C6H10O5−H] , 255.0651, 153.0187 [M−C6H10O5−CO−C7H6O2−H] , 135.0082 [M−C6H10O5−CO−C7H6O2−H2O−H] , 119.0497, 91.0184

67.5

[36]

27

Quercetin-3β-D-glucoside

14.55

C21H20O12

464.0936

0.21

463.0871 [M-H]

463.0872 [M-H]

301.0338 [M−H−C6H10O5] , 300.0270, 271.0247

[29]

28

2- [1-(2H-1,3-Benzodiox ol- 5-yl) propan-2-yl]-6-metho xy-4-(prop-2-en-1-yl) phenol

14.74

C20H22O4

326.1516

0.63

327.1590 [M+H] +

327.1593 [M+H] +

312.1348, 295.1328 [M−OCH2+H] +, 280.1095, 263.1071, 251.0001, 235.1122, 175.0758, 163.0753 [M−C10H12O2+H] +, 151.075, 137.0596, 133.0647, 103.0540, 98.9841

71.6

29

Cinnamylalcohol-6'-O-α-furanara-

binose-O-β-glucopyranoside

14.81

C20H28O10

428.1664

−4.38

427.1598 [M-H]

427.1582 [M-H]

293.0861, 233.0650, 191.0549, 161.0451, 149.0448, 125.0239, 89.0240, 81.0344, 59.0136

85.9

[42]

CR

30

6-Methoxymellein

14.99

C11H12O4

208.0734

−0.55

209.0804 [M+H] +

209.0802 [M+H] +

191.0701, 181.0847, 177.0544, 163.0765, 149.0596 [M−COOCH+H] +, 131.0486, 121.0647 [M−2OH−OCH3−CO+H] +, 109.0647, 103.0540, 93.0698, 91.0540, 55.0177

85.8

.

31

Coumarin

15.18

C9H6O2

146.0366

−1.33

147.0441 [M+H] +

147.0439 [M+H] +

127.0543, 103.0541 [M−CO2+H] +, 91.0540 [M−2CO+H] +, 43.0242

96.8

Standard,

[42]

CR, CC, C. cassia leaves

32

Quercetin

15.36

C15H10O7

302.0424

−0.98

303.0499 [M+H] +

303.0496 [M+H] +

303.0496, 275.0399 [M−CO+H] +, 257.0446 [M−CO−H2O+H] +, 247.0590, 229.0491 [M−2CO−H2O+H] +, 199.0434 [M−3CO−H2O+H] +165.0178, 163.0389,

153.0183 [M−CO−C7H6O2+H] +, 133.0231[M−CO−C7H4O2−H2O+H] +, 121.0297, 111.0075

99.8

[24], [30]

CR

33

Quercitrin

15.36

C21H20O11

448.1003

−0.58

449.1089 [M+H] +

449.1087 [M+H] +

431.0983, 369.0594, 345.0606, 315.0494, 303.0497 [M+H−C6H9O4] +, 257.0439 [M−C6H9O4−CO−H2O+H] +, 229.0492 [M−C6H9O4−2CO−H2O+H] +, 129.0548, 85.0283, 71.0490

82.8

Standard

34

Graveobioside A

15.36

C26H28O15

580.1402

−4.46

579.1344 [M-H]

579.1325 [M-H]

476.1080, 417.1531 [M−C6H10O5−H] , 300.0259 [M−C6H8O4−C5H10O4−H] , 271.0235 [M−C11H16O10−H] , 178.9979 [M−C17H20O11−H]

85.3

35

Libertellenone B

15.74

C20H26O4

330.1830

−0.32

331.1903 [M+H] +

331.1902 [M+H] +

313.1796 [M−H2O+H] +, 295.1676, 271.1686, 243.1763, 165.0911, 125.0565 [M−C12H14O3+H] +

75.2

36

Yucalexin P-17

16.06

C17H20O3

272.1412

0.27

273.1485 [M+H] +

273.1488 [M+H] +

255.1391 [M−H2O+H] +, 245.1534, 227.1441 [M−H2O−CO+H] +, 203.1070, 149.0964 [M−CH3−C6H6O2+H] +, 82.8045 [M−C11H10O3+H] +

86.6

37

Azelaic acid

16.24

C9H16O4

188.1040

0.12

187.0965 [M-H]

187.0966 [M-H]

169.0861, 143.1072, 125.0966 [M−COOH−OH−H] , 123.0811, 97.0654 [M−2COOH−H] , 57.0343

95.4

[43], [44]

CR

38

Kaempferol

16.27

C15H10O6

286.0475

−0.89

287.0550 [M+H] +

287.0548 [M+H] +

287.0548, 258.0511 [M−CO+H] +, 183.0288, 165.0183, 153.0189 [M−CO−C7H6O+H] +, 133.0292,121.0281

99.2

[30]

CR, CC, C. cassia leaves

39

1-(Carboxymethyl) cyclohexane carboxylic acid

16.36

C9H14O4

186.0884

1.06

185.0808 [M-H]

185.0819 [M-H]

141.0916 [M−COO−H] , 104.0775 [M−C6H9−H]

87.2

40

Kaempferol-3-O-α-L-arabinopyranosyl-7-O-α-L-rhamnopyranoside

16.64

C26H28O14

564.1456

−4.12

563.1393 [M-H]

563.1384 [M-H]

435.2045, 285.0416 [M−C6H8O4−C5H10O4−H] , 284.0316, 255.0286, 147.5166, 70.7867

[45]

41

2-Methoxy benzaldehyde

16.67

C8H8O2

136.0522

−0.16

137.0597 [M+H] +

137.0596 [M+H] +

109.0647, 107.0490 [M−OCH2+H] +, 93.0698 [M−CO−CH3+H] +, 81.0697, 79.0512 [M−CO−OCH3+H] +

90.9

[23]

CR, C. cassia leaves

42

Cinnamyl alcohol

17.22

C9H10O

134.0726

−0.89

117.0698

[M+H-H2O] +

117.0696 [M+H-H2O] +

117.0696 [M−H2O+H] +, 91.0540 [M−C2H4O+H] +, 78.2648 [M−C3H5O+H] +, 63.4672, 49.4958

standard

CR, CC, C. cassia leaves

43

4-Methylumbelliferyl-α-D-glucopyranoside

17.37

C16H18O8

338.1002

0.21

339.1074 [M+H] +

339.1075 [M+H] +

321.0970 [M−OH+H] +,177.0546 [M−C6H10O5+H] + ,145.0284 [M−C6H10O5−CH3−OH+H] +, 127.0389 [M−C6H10O5−CH3−OH+H] +, 97.0280

94.6

44

(±)-Abscisic acid

17.55

C15H20O4

264.1359

−0.85

265.1484 [M+H] +

265.1481 [M+H] +

247.1332 [M−H2O+H] +,, 229.1216 [M−2H2O+H] +,187.1108 [M−O−CH2−COO+H] +

90.2

45

trans-Cinnamic acid

18.16

C9H8O2

148.0518

−0.60

149.0232 [M+H] +

149.0231 [M+H] +

144.9817, 131.0493 [M−H2O+H] +, 121.0282, 116.9669, 107.0491, 105.0539 [M−CO+H] +, 93.0698, 79.0545

Standard

[23],[25]

CR, CC, C. cassia leaves

46

4-Phenyl-3-buten-2-one

18.31

C10H10O

146.0730

−1.03

147.0803 [M+H] +

147.0803 [M+H] +

132.0567 [M−CH3+H] +, 129.0699, 119.0854, 117.0698, 107.0489 [M−CH−CO+H] +, 91.0541 [M−C3H4O+H] +, 79.0542

95.3

[46], [47]

C. verum

47

3-Tert-butyladipic acid

18.32

C10H18O4

202.1196

−1.36

201.1121 [M-H]

201.1120 [M-H]

183.1021 [M−OH−H] , 156.8982 [M−COO−H] , 139.1124

70.0

[25]

48

trans-Cinnamaldehyde

18.84

C9H8O

132.0573

−0.19

133.0647 [M+H] +

133.0646 [M+H] +

115.0540 [M−H2O+H] +, 105.0697 [M−CO+H] +, 103.0542, 91.0541 [M−CO−CH2+H] +, 79.0542 [M−CO−C2H2+H] +, 55.0178 [M−C6H6+H] +

97.9

Standard,

[25]

CR, CC

49

2-Methoxycinnamic acid

19.54

C10H10O3

178.0629

−0.51

161.0597

[M+H-H2O] +

161.0596 [M+H-H2O] +

146.0366,133.1011 [M−H2O−CO+H] +, 119.0855 [M−CHO−OCH3+H] +, 105.0698 [M−COOH−CH+H] +, 91.0544 [M−CO−CH2−OCH3+H] +

Standard,

[25]

CR, CC, C. cassia leaves

50

9S,13R-12-Oxophytodienoic acid

20.08

C18H28O3

292.2037

−0.68

293.2111 [M+H] +

293.2104 [M+H] +

275.2003 [M−H2O+H] +, 257.1893, 239.1799[M−C4H6+H] +, 229.1953, 163.1117, 159.1167, 147.1163[M−C7H14+H] +, 133.1012, 107.0855, 95.0853, 81.0698 [M−C12H20O3+H] +, 69.0699

91.0

[48]

51

Corchorifatty acid F

20.43

C18H32O5

328.2237

−0.10

327.2166 [M-H]

327.2165 [M-H]

309.2062, 291.1955, 242.9845 [M−C5H4−OH−H] , 239.1283, 229.1435, 221.1171, 211.1313, 185.1173, 183.1374, 171.101 [M−C9H16O2−H] , 137.0968, 97.0655, 85.0290 [M−C13H22O4−H] , 57.0343

[29], [49]

52

Deoxyphomalone

20.47

C13H18O4

238.1204

−0.29

239.1277 [M+H] +

239.1275 [M+H] +

221.1171, 205.1192 [M−2OH+H] +, 179.0705 [M−C2H5−OCH3+H] +, 174.0678, 163.0750, 151.0753 [M−C2H5−C3H7O+H] +, 137.0598 [M−2OH−2OCH3−C2H4−C3H3+H] + , 135.0799, 107.0481, 95.0861 [M−OH−2OCH3−C2H5−C4H3O+H] +, 59.0490

74.4

53

4-Ethylbenzaldehyde

20.73

C9H10O

134.0730

−0.14

135.0804 [M+H] +

135.0803 [M+H] +

120.0567, 107.0490 [M−CO+H] +, 105.0697 [M−C2H6+H] +, 103.0542, 79.0542 [M−C2H6−CO+H] +

92

[50]

CR

54

1-Naphthol

21.00

C10H8O

144.0573

−0.09

145.0648 [M+H] +

145.0647 [M+H] +

116.0575 [M−C−OH+H] +, 115.0541, 102.0468 [M−C2H2−OH+H] +, 91.0539 [M−C3H2−OH+H] +, 84.9598

89

[51], [52]

CR

55

4-Methoxy cinnamaldehyde

21.02

C10H10O2

162.0679

0.04

163.0753 [M+H] +

163.0754 [M+H] +

145.0650, 135.0805 [M−CO+H] +, 133.0648, 110.0203 [M−C3H3O+H] + , 107.0491, 105.0699 [M−CO−OCH3+H] +, 91.0542, 79.0542 [M−C3H3O− OCH3+H] +, 55.0178

88.4

[42]

CR, CC

56

9,12,13-Trihydroxy-15-octadecenoic acid

21.72

C18H34O5

330.2393

−0.05

329.2322 [M-H]

329.2322 [M-H]

311.2227 [M−H2O−H] , 293.2102 [M−2H2O−H] , 229.1433, 211.1331, 183.1383, 171.1018, 139.1123, 127.1120, 125.0975, 99.0812, 57.0342

90.0

57

(−)-Caryophyllene oxide

22.32

C15H24O

220.1826

−0.53

221.1899 [M+H] +

221.1900 [M+H] +

203.1795, 175.1483 [M−O−2CH2−C+H] +, 161.1323 [M−2CH3−CO−CH+H] +, 147.1169 [M−2CH3−CO−CH−CH2+H] +, 133.1010, 119.0855, 95.0855

92.9

[53]

CR, CC, C. cassia leaves

58

4-Methoxychalcone

28.61

C16H14O2

238.0992

0.66

239.1066 [M+H] +

239.1073 [M+H] +

221.0961, 193.1012, 178.0875, 161.0595 [M−C6H6+H] +,133.0647 [M−C7H6O+H] +, 115.054, 105.0333 [M−C6H6−C2H−OCH3+H] +

86.7

[54]

C. cassia

a indicated that the comprehensive score of molecular formula, molecular structure and MS2 fragment ions matching with the mzCloud database.

b indicated that this compound has been isolated or identified from a certain plant.

Reviewer 3 Report

The corrections made by the authors are satisfactory.

Author Response

Dear reviewer 3,

Thank you again for the helpful advice and affirmation.

Best wishes.

Yours sincerely,

Li-ping Dai